# FreRA: A Frequency-Refined Augmentation for Contrastive Learning on Time Series Classification

## Abstract

Contrastive learning has emerged as a competent approach for unsupervised representation learning. However, the design of an optimal augmentation strategy, although crucial for contrastive learning, is less explored for time series classification tasks. Existing predefined time-domain augmentation methods are primarily adopted from vision and are not specific to time series data. Consequently, this cross-modality incompatibility may distort the global semantics of time series by introducing mismatched patterns into the data. To address this limitation, we present a novel perspective from the frequency domain and identify three advantages for downstream classification: 1) the frequency component naturally encodes *global* features, 2) the orthogonal nature of the Fourier basis allows *easier isolation* and *independent modifications* of critical and unimportant information, and 3) a *compact* set of frequency components can preserve semantic integrity. To fully utilize the three properties, we propose the lightweight yet effective **Fre**quency-**R**efined **A**ugmentation (**FreRA**) tailored for time series contrastive learning on classification tasks, which can be seamlessly integrated with contrastive learning frameworks in a plug-and-play manner. Specifically, FreRA automatically separates critical and unimportant frequency components. Accordingly, we propose Identity Modification and Self-adaptive Modification to protect global semantics in the critical frequency components and infuse variance to the unimportant ones respectively. Theoretically, we prove that FreRA generates semantic-preserving views. Empirically, we conduct extensive experiments on two benchmark datasets including UCR and UEA archives, as well as 5 large-scale datasets on diverse applications. FreRA consistently outperforms 10 leading baselines on time series classification, anomaly detection, and transfer learning tasks, demonstrating superior capabilities in contrastive representation learning and generalization in transfer learning scenarios across diverse datasets.

## 1 Introduction

Time series classification has been an essential problem in a wide range of applications, such as activity recognition (Qian et al., 2019), speech recognition (Huijben et al., 2023), and industrial monitoring (Eldele et al., 2023). Despite the promising performance achieved by supervised methods (Qian et al., 2019), a large number of accurate labels are required to deliver good performance. However, label annotation for time series without human error is costly and time-consuming. This is because time series data are not intuitively recognizable or meaningful for humans, unlike images or language. Given the circumstance, contrastive learning has been attested as a compelling framework for representation learning in the absence of labels (Meng et al., 2023b; Qian et al., 2022). Specifically, it learns to solve an instance discrimination pretext task (Wu et al., 2018) that aims to distinguish different samples (negative pairs) while keeping different views of the same sample (positive pairs) close, wherein different views are usually generated by a set of augmentation functions.

Despite the prevalence of contrastive learning (Chen & He, 2021; Huang et al., 2023), its efficacy heavily relies on the proper selection of data augmentation (Luo et al., 2023; Tian et al., 2020). Existing works in time series contrastive learning often apply carefully hand-picked transformations such as `jitter-and-scale` and `permutation-and-jitter` (Eldele et al., 2023). These

augmentations are mostly adopted from the vision domain and do not take the intrinsic characteristics of time series into consideration. Due to the unintuitive nature of time series, it becomes impractical to painlessly figure out semantically compromised augmented samples, unlike in vision. As a result, when applying predefined augmentation, the type and degree of transformation need to be carefully selected to reduce the loss of semantic information. Trials and errors for hand-picked augmentation make it costly to apply. What's worse is that there is no single augmentation function that consistently performs well on all diverse datasets (Qian et al., 2022). As a result, recent works have started to explore the generalized principle and design of transformation that produce universally optimal augmentation $v^*$ for time series contrastive learning. For instance, the latest InfoTS (Luo et al., 2023) and AutoTCL (Zheng et al., 2024) share a common principle that optimal augmentation should remain semantically consistent with their anchor samples $MI(v^*; y) = MI(x; y)$, where $x$ and $y$ are the random variable denoting time series sample and label, and $MI(\cdot; \cdot)$ represents the mutual information (MI) quantifying the mutual dependence between two variables. However, we find out that empirically these proposed augmentation strategies still fail to preserve semantic integrity. To be more specific, they more or less undermine or disrupt the meaningful patterns with respect to the global semantics, i.e., $MI(v^*; y)$, of the time series , which will be discussed in detail in the later sections. The global semantics, whose amount is quantified as $MI(x; y)$ in our analysis, refer to *the information that spans the entire time series and contributes significantly to distinguishing between different classes*. Therefore, it is crucial for view generation in contrastive learning on time classification tasks.

For a clearer illustration, we plot 6 different $MI(v; y)$ curves in Figure 1, where $v \in \{\mathcal{A}_s(x), x, \mathcal{A}_{AutoTCL}(x), \mathcal{A}_{InfoTS}(x), \mathcal{T}_T(x), \mathcal{T}_F(x)\}$, denoting the augmented view generated by our proposed FreRA, identity transformation, AutoTCL, InfoTS and `jitter-and-scale` and `amplitude-and-phase-perturbation`, respectively, and $y$ is the label in the downstream classification task. The $x$-axis presents the timestamp of the time series, and the $y$-axis denotes the value of MI. We provide more details regarding Figure 1 in Appendix A.1. Intuitively, a higher MI curve is preferable because it indicates more global semantic information is preserved. We first observe the proposed FreRA (blue curve) achieves the highest value among all the curves, and it almost overlaps with $MI(x; y)$ (orange curve), indicating FreRA preserves all the semantic information in the generated views and there is no major loss of critical information. We then observe the other 3 curves are consistently lower than the first two curves, indicating the semantics are undermined in the latter three transformations, which agrees with our earlier analysis. Previous work (Xu et al., 2024) figures out that undermined semantics in the views cause degraded representation and harm the performance of downstream tasks, which is undesirable.

Despite strong empirical performance on certain datasets, existing augmentations undermine global semantics, which reminds us of the limitations of time-domain augmentations. Due to the *inter-correlation among timestamps*, time-domain manipulations *fail to keep the critical global information intact while introducing variation*. To overcome such limitations , we present a novel perspective from the frequency domain, more specifically, frequency refinement. We first identify 3 advantages of the frequency

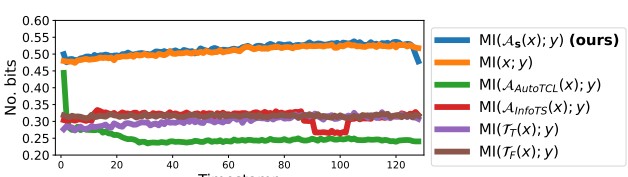

Figure 1: Our method (blue curve) achieves the highest MI between the views generated and the label, enabling better semantic preservation compared with SOTA. The global semantics are well preserved to facilitate contrastive representation learning.

domain over the time domain: 1) *global*: each frequency component encapsulates a global feature that spans all timestamps and is more meaningful in revealing the global semantics for classification tasks; 2) *independent*: the orthogonal Fourier basis ensures the independence among frequency components, making it unlikely to contain both critical and noisy information at the same time, which allows clear separation and independent manipulations on different components; and 3) *compact*: given the first two properties, there is a compactly distributed set of frequency components that can well preserve the semantic integrity. The three advantages of the frequency domain and how they facilitate the design of FreRA will be elaborated in detail in the latter sections.

To fully tap into the great potential of the frequency domain, we propose a novel **Fre**quency-**R**efined **A**ugmentation (**FreRA**) for contrastive learning on time series classification. The central idea of FreRA is to adaptively refine frequency components. Specifically, we learn a lightweight trainable parameter vector to capture the inherent semantic distribution in the frequency domain. Identity modification and self-adaptive modification are then proposed to the well-separated critical and unimportant frequency components respectively, to preserve semantics and infuse variance. This single-parameter vector adeptly guides the refinement in both the separation and modifications. FreRA is a generalized transformation that automatically adapts to training data, alleviating manual efforts in adjusting augmentations. It also ensures that the added variation does not compromise the global semantics by refining the frequency domain rather than the time domain. FreRA can be easily adapted to a wide range of contrastive learning models in a plug-and-play manner. In summary, our main contributions are:

- We identify three advantages of the frequency domain and introduce the novel frequency perspective to automatic view generation for time series contrastive learning for the first time.

- Building upon these advantages, we design a lightweight and unified automatic augmentation FreRA for contrastive representation learning on classification tasks, which can be applied in a plug-and-play manner and jointly optimized with the contrastive learning model.

- Extensive experiments on 135 benchmark datasets demonstrate the competitive performance of FreRA in contrastive learning and improved generalization in transfer learning scenarios on both time series classification and anomaly detection tasks.

## 2 RELATED WORK

**Time Series Contrastive Learning.**    Considering the challenges of data annotation for time series, contrastive learning achieves great success in time series applications (Yue et al., 2022b; Tonekaboni et al., 2021; Eldele et al., 2023; Meng et al., 2023a). TS2Vec (Yue et al., 2022b) performs hierarchical contrastive learning to learn timestamp-wise representations. TNC (Tonekaboni et al., 2021) learns temporal representations where neighboring and non-neighboring signals are distinguishable. TS-TCC (Eldele et al., 2023) proposes a novel cross-view prediction task. MHCCL (Meng et al., 2023a) utilizes hierarchical clustering for temporal contrastive representation learning. Although previous works introduce various architectures and objectives, the essence of contrastive learning lies in the attraction of positive pairs and the repulsion of negative pairs (He et al., 2020), making view generation a crucial component.

**Frequency Domain of Time Series.**    The frequency domain mostly serves as a substitute or supplementary modality in multiple time-series tasks, e.g., representation learning (Yang & Hong, 2022), domain generalization (Zhang et al., 2022), and time series forecasting (Zhou et al., 2022a;b; Yi et al., 2023). Those works empirically discover and exploit the frequency domain as an informative element: BTSF and TF-C (Yang & Hong, 2022; Zhang et al., 2022) encourage time-frequency consistency in representation learning to enhance generalization; Zhou et al. (2022a) claim that utilizing low-frequency Fourier components for time series forecasting could undermine noise; Zhou et al. (2022b) prove that a subset of randomly selected Fourier components preserves most of the information in the time series. Yi et al. (2023) find that the frequency domain possessed a global view and compact energy in MLP-based time series forecasting. Those works provide heterogeneous understandings of identifying essential information in the frequency domain, either from domain knowledge or heuristics. In contrast, our motivation inspires a unified approach that manipulates frequency-domain information to facilitate contrastive learning. Wen et al. (2020) explore frequency-domain transformations for training data enhancement on supervised time series tasks. However, our focus is on the contrastive learning setting, where the methods therein are not readily applicable.

**Augmentations for Contrastive Learning.**    As a crucial component for contrastive learning, augmentation functions are either carefully designed or selected from grid search (Qian et al., 2022; Eldele et al., 2023). The former requires domain knowledge, while the latter is computationally inefficient. There is no single existing augmentation function enjoying universal optimal performance (Qian et al., 2022). The selection is task-dependent (Tian et al., 2020) and subject to data modality (Jaiswal et al., 2020). Some works try to automate the selection from a predefined set of

transformations or adapt a well-defined transformation to serve contrastive learning: InfoTS (Luo et al., 2023) trains a data-driven probabilistic augmentation selector that intends to encourage high fidelity and variety to select optimal augmentation. Demirel & Holz (2024) introduce tailored mixup for non-stationary quasi-periodic time series. Another line of work eliminates the use of hand-designed augmentation: InfoMin (Tian et al., 2020) generates contrastive views with a flow-based model, guided by the adversarial InfoMin objective. AutoTCL (Zheng et al., 2024) factorizes the time series instance to informative and noisy parts by timestamps. Self-adaptive augmentation in the frequency domain is less explored in contrastive learning, and we fill this research gap in this work.

## 3 METHODOLOGY

### 3.1 PROBLEM STATEMENT

Let $x = [x^1, x^2, ..., x^L]^T \in \mathbb{R}^{L \times D}$ denote an unlabeled time series instance that lasts for $L$ timestamps and has $D$ channels where the signal at $i$-th timestamp $x^i \in \mathbb{R}^D, \forall i \in [1, ..., L]$. We do not make assumptions about the dimension or length of the time series. Our problem definition is valid for both univariate and multivariate time series datasets of varying scales. $\mathcal{F}(\cdot)$ and $\mathcal{F}^{-1}(\cdot)$ represent the Fourier transform and its inverse, respectively. We denote $x_f = [x_f^1, x_f^2, ..., x_f^F]^T = \mathcal{F}(x) \in \mathbb{C}^{F \times D}$ as the Fourier transform of $x$, i.e., $x = \mathcal{F}^{-1}(\mathcal{F}(x))$, where $\mathbb{C}$ stands for the complex space and $F = \lfloor L/2 \rfloor + 1$ is the number of frequency components. $x_f^1$ and $x_f^F$ embed the characteristics of the lowest and highest Fourier frequency basis functions, respectively.

In the general contrastive learning framework, an encoder $f_\theta$ is trained to map input samples to a latent space where the downstream task is performed. Taking SimCLR (Chen et al., 2020) as our contrastive learning framework, it appends a projector $g_\phi$ to the encoder. $\theta$ and $\phi$ denote the sets of trainable parameters in the encoder and projector respectively. In the mini-batch $X \in \mathbb{R}^{B \times L \times D}$ containing $B$ instances, each anchor $x \in X$ associates with its augmented view $\mathcal{A}(x)$ as a positive pair, and with the other $(B - 1)$ samples to form negative pairs. We consider the batch-wise contrastive loss as: $\mathcal{L}_{\text{CL}} = \mathcal{L}(X; \mathcal{A}(\cdot), f_\theta, g_\phi)$, which will be elaborated later.

It is a common belief in existing works (Tian et al., 2020; Luo et al., 2023; Zheng et al., 2024) that the optimal view generator for contrastive learning is defined as follows.

**Definition 1** (**Optimal View Generator**). *Given the random variable $x$ denoting the input instances, its optimal view generator $A^*(\cdot)$ should satisfy $\mathcal{A}^*(x) = \arg\min_{\mathcal{A}} MI(\mathcal{A}(x); x)$, subject to*

$MI(\mathcal{A}^*(x); y) = MI(x; y)$.

Based on the definition, an optimal view generator should preserve the *minimal but sufficient* information with respect to the semantics of its input. Existing works on time series contrastive learning mainly select an empirically optimal augmentation function $\mathcal{T}^*(\cdot)$ from a set of predefined transformations $\{\mathcal{T}_1(\cdot), \mathcal{T}_2(\cdot), ..., \mathcal{T}_m(\cdot)\}$, such as $\{\texttt{scaling}, \texttt{jittering}, \texttt{rotation}\}$, i.e., $\mathcal{T}^*(\cdot), \theta^*, \phi^* = \arg\min_{\mathcal{T}_i(\cdot) \in \{\mathcal{T}_1(\cdot), \mathcal{T}_2(\cdot), ..., \mathcal{T}_m(\cdot)\}, \theta, \phi} \mathcal{L}(X; \mathcal{T}_i(\cdot), f_\theta, g_\phi)$. Selected from the painstaking trials and errors, $\mathcal{T}^*(\cdot)$ still suffers from loss of semantic information. Other works utilize a trainable network to model the transformation function, denoted as $\mathcal{T}(\cdot; \gamma)$, where $\gamma$ is the parameters of the transformation network. They optimize the entire framework as follows: $\gamma^*, \theta^*, \phi^* = \arg\max_{\gamma} \arg\min_{\theta, \phi} \mathcal{L}(X; \mathcal{T}(\cdot; \gamma), f_\theta, g_\phi) + \mathcal{L}_{\text{auxiliary}}(\gamma)$, where $\mathcal{T}(\cdot; \gamma^*)$ is the learned transformation function and $\mathcal{L}_{\text{auxiliary}}(\gamma)$ is the extra regularization on the transformation network. The optimization for the min-max objective is done through an alternative update of the transformation network and the contrastive learning model.

Aware of the selection cost, compromised semantics, and the complex alternative optimization in previous approaches, we aim to develop a semantic-preserving automatic augmentation $\mathcal{A}(\cdot)$ that can be jointly optimized with the contrastive learning model, with objective formulated as follows:

$$\arg\min_{\mathcal{A}(\cdot), \theta, \phi} \mathcal{L}(X; \mathcal{A}(\cdot), \theta, \phi) + \mathcal{L}_{\text{auxiliary}}(\mathcal{A}(\cdot)) \tag{1}$$

subject to $\text{MI}(\mathcal{A}(x); y) = \text{MI}(x; y)$.

Figure 2: An overview of the proposed FreRA. The left-hand side presents the detailed design of FreRA: identity modification on critical components and self-adaptive modification on unimportant components are conducted in the frequency domain to maintain contextual information and infuse variance respectively. The matching colors between $\mathbf{s}$ and $\mathbf{w}_{\text{dist}}$ on unimportant components intend to illustrate the adaptive distortion. The independent manipulations in FreRA ensure the added variance does not impact the critical semantic information. As a plug-and-play component, FreRA can be jointly trained with any contrastive learning framework, as illustrated on the right-hand side. The contrastive learning model is pre-trained in the time domain. FreRA encourages the compactness of critical frequency components and the consistency of positive pairs' representations. In evaluation, a classifier is trained on top of the frozen pre-trained encoder to get predictions for downstream tasks.

In the following section, we will present the design of $\mathcal{A}(\cdot)$ and prove the fulfillment of Definition 1 as well as the criterion in the objective.

## 3.2 OVERVIEW OF FRERA

It is a common belief that a good view in contrastive learning should contain both semantic-preserving information and a considerable amount of variance (Zheng et al., 2024; Luo et al., 2023). The former ensures strong performance on downstream tasks, while the latter encourages the encoder to learn generalizable representations. To achieve such a good view, we leverage the global, independent, and compact properties of the frequency domain to design the frequency-refined augmentation, FreRA as follows:

$$\mathcal{A}_{\mathbf{s}}(x) = \mathcal{F}^{-1}( \underbrace{\mathbf{w}_{\text{crit}} \odot x_f}_{\text{global and compact}} + \underbrace{\mathbf{w}_{\text{dist}} \odot x_f}_{\text{independent}} ) \in \mathbb{R}^{L \times D}, \tag{2}$$

where $\odot$ denotes elementwise multiplication and $\mathbf{s}$ is the lightweight trainable parameter of FreRA. Specifically, $\mathbf{w}_{\text{crit}} = [w_{\text{crit}}^1, w_{\text{crit}}^2, ..., w_{\text{crit}}^F] \in \{0, 1\}^F$ applies identity modification on those identified critical frequency components to preserve the essential information, while $\mathbf{w}_{\text{dist}} \in \mathbb{R}_{\geq 0}^F$ applies self-adaptive modification to the unimportant frequency components to introduce diverse distortion. The two modifications are applied independently to keep the critical global information intact while introducing variance. There may exist certain component $x_f^i$ whose $w_{\text{crit}}^i$ and $w_{\text{dist}}^i$ are both 0. The refinement, including the component separation and modifications, is guided by a single vector $\mathbf{s}$. Figure 2 depicts an overview of the proposed FreRA.

### 3.2.1 WHY FRERA MAKES GOOD VIEWS?

In this section, we elaborate the three advantages of the frequency domain over the time domain and elaborate on them in detail. Based on them, we explain why the frequency-domain refinement produces good views that benefit contrastive representation learning for downstream tasks. To facilitate our analysis, we introduce a new set of notation for time-domain data $x(n)$, augmented view $x_{\mathcal{A}}(n)$, and frequency-domain data $X(m)$ as follows:

$$x(n) = x^{n+1}, \quad x_{\mathcal{A}}(n) = \mathcal{A}_{\mathbf{s}}(x^{n+1}) \quad \text{for } n \in \{0, 1, ..., L-1\},$$

$$X(m) = \begin{cases} x_f^{m+1}, & \text{if } m+1 \leq F \\ \overline{X(L-m)} = \overline{x_f^{L-m+1}}, & \text{otherwise,} \end{cases} \quad \text{for } m \in \{0, 1, ..., L-1\} \tag{3}$$

where $\overline{x_f^{L-m+1}}$ is the conjugate of $x_f^{L-m+1}$, $x = [x(0), x(1), ..., x(L-1)]^T$ and $x_f = [X(0), X(1), ..., X(F-1)]^T$. We present the derivation for the second condition of $X(m)$ in Appendix A.3.1.

**Global.** The Fourier component is derived by Discrete Fourier Transform (DFT) (Sundararajan, 2001): $X(m) = \sum_{n=0}^{L-1} x(n) e^{-\frac{2\pi i}{L} mn}$, where each frequency component $X(m)$ encodes all the timestamps. According to the Dual convolution theorem (Sundararajan, 2001), element-wise multiplication in the frequency domain is equivalent to circular convolution in the time domain. Then we have $\mathcal{F}(\tilde{\mathbf{w}} * x) = \frac{1}{F} \mathcal{F}(\tilde{\mathbf{w}}) \odot \mathcal{F}(x)$, where $*$ denotes the circular convolution operator. Let $\mathcal{F}(\tilde{\mathbf{w}}) = \mathbf{w}_{\text{crit}}$, we can conclude that the frequency modification is equivalent to time-domain convolution with kernel $\tilde{\mathbf{w}} = \mathcal{F}^{-1}(\frac{1}{F} \mathbf{w}_{\text{crit}}) \in \mathbb{C}^L$, which has global receptive field on $x$. This global perspective is crucial to the time series classification tasks, as it *preserves global semantics across the entire time series and ensures that all timestamps are altered with distortion applied only to unimportant components.*

**Independent.** The inverse DFT, $x(n) = \frac{1}{L} \sum_{m=0}^{L-1} X(m) e^{\frac{2\pi i}{L} mn}$, offers an alternative perspective of interpreting the frequency components: they are the coefficients of the orthogonal decomposition of the time domain. The decomposition basis for $X(m)$ is $\mathbf{u}_m = [e^{\frac{2\pi i}{L} mn} | n = 0, 1, ..., L-1]^T \in \mathbb{C}^L$. We have $\langle \mathbf{u}_m, \mathbf{u}_q \rangle = 0$ if $m \neq q$, where $\langle \mathbf{u}, \mathbf{v} \rangle = \mathbf{u}^T \overline{\mathbf{v}} \in \mathbb{C}$ is the Hermitian inner product. The proof is presented in Appendix A.3.2. The zero-valued Hermitian inner product confirms the orthogonal nature of the decomposition basis. Each coefficient $X(m)$ measures the contribution of its corresponding basis function independently. Similarly, when FreRA modifies frequency components, each modified components independently contribute to the augmented views without being affected by the others. The independence *makes it easy to isolate critical and unimportant information* by updating $\mathbf{w}_{\text{crit}}$ and $\mathbf{w}_{\text{dist}}$ and *prevent added variance from degrading critical information.*

**Compact.** Parseval's theorem (Parseval, 1806) states that the total energy of the signal in the time domain is equal to the average energy in the frequency domain, i.e., $\sum_{n=0}^{L-1} |x(n)|^2 = \frac{1}{L} \sum_{m=0}^{L-1} |X(m)|^2$. This implies that if most energy is concentrated in a small number of frequency components, the information of the signal is compactly distributed in the frequency domain. Figure 3 in the Appendix validates this interpretation by showing that most energy is concentrated on the first ten frequency components for the UCIHAR dataset and the same principle holds for other datasets. This aligns with our common sense that many natural or man-made processes recorded as time series encode information in low-frequency components. However, for classification tasks, the semantically relevant bandwidth is normally unknown and the importance of these components varies, making it hard to automatically rank their contributions and identify critical ones. Moreover, the exception happens in certain applications, such as audio processing (Virtanen et al., 2015), where both low- and high-frequency components matter. As critical components that encapsulate the semantic meaning of the signal are likely a subset of the compactly distributed informative components, their distribution should also remain compact. This leads us to *enforce compactness in identifying the critical components in the frequency domain*. Notably, the range of the energy in Figure 3 highlights the shared distribution of the informative components in the frequency domain. It advocates that a single vector $\mathbf{s}$ is sufficient to work across all the samples in the dataset.

Lastly, we demonstrate that FreRA preserves global semantics, i.e., $\text{MI}(\mathcal{A}_{\mathbf{s}}(\mathsf{x}); \mathsf{y}) = \text{MI}(\mathsf{x}; \mathsf{y})$ (Proposition 3 in the Appendix with proof) under the reliable assumption that noisy frequency components are independent to the label. This proposition agrees with our observation in Figure 1 where the blue and orange curves nearly overlap. It also shows that FreRA satisfies the semantic-preserving constraint in Definition 1, leaving only the minimization objective for optimization.

### 3.3 TIME SERIES CONTRASTIVE LEARNING WITH FRERA

In this section, we first elaborate on the detailed design of FreRA and propose the objective that allows the joint training of FreRA and the contrastive learning framework.

**Discern the Importance of Frequency Components**  Both $\mathbf{w}_{\text{crit}}$ and $\mathbf{w}_{\text{dist}}$ are parameterized by a lightweight trainable vector $\mathbf{s} = [s_1, s_2, ..., s_F]^T \in \mathbb{R}^F$, where $s_i$ scores the importance of the $i$-th frequency component $x_f^i$ for the global semantics. A higher $s_i$ indicates the contextual importance of $x_f^i$. On the other hand, $s_i$ with a negative value suggests $x_f^i$ is the noise component.

**Identity Modification on Critical Frequency Components.** A simple way to derive a binary vector like $\mathbf{w}_{\text{crit}}$ is to sample from a Bernoulli distribution controlled by the parameter $\mathbf{p} = [p_1, p_2, ..., p_F]^T \in \mathbb{R}^F$, i.e., $w_{\text{crit}}^i \sim \text{Bernoulli}(p_i)$ for $i \in [1, 2, ..., F]$, where $p_i$ denotes the probability that the $i$-th frequency component is semantically critical. Meanwhile, the Bernoulli distribution is not differentiable w.r.t. $p_i$. Instead, we apply the Gumbel-Softmax reparameterization (Jang et al., 2017), i.e., $w_{\text{crit}}^i = \text{Gumbel-Softmax}(p_i)$. The importance score vector $\mathbf{s}$ makes it possible because its values can be used to reflect the probability, i.e., $p_i = \sigma(s_i)$, where $\sigma(\cdot)$ is the sigmoid function. The reparameterization is formulated as follows:

$$w_{\text{crit}}^i = \sigma((\log \epsilon - \log(1 - \epsilon) + \log \frac{\sigma(s_i)}{1 - \sigma(s_i)})/\tau_w), \tag{4}$$

where $\epsilon \sim \text{Uniform}(0, 1)$ and $\tau_w$ is the temperature controlling the discretization. As $\tau_w \to 0$, $w_{\text{crit}}^i$ approximates a Bernoulli distribution: $P(w_{\text{crit}}^i \to 0) = 1 - p_i$ if $\epsilon > p_i$, and $P(w_{\text{crit}}^i \to 1) = p_i$ if $\epsilon < p_i$. In this way, distinct importance score $s_i$ is assigned to $x_f^i$ to capture varying levels of contextual relevance within each frequency component.

**Self-adaptive Modification on Unimportant Frequency Components.** Besides preserving contextually relevant information, a good view also requires variance to be infused. Instead of adding random noise, we deliberately modify the unimportant noisy components identified by $\mathbf{s}$ to avoid affecting critical information. As the score $s_i$ indicates the importance of the $i$-th frequency component $x_f^i$ for global semantics, frequency components with smaller values are considered unimportant. A threshold value is required to separate the unimportant components from the rest and handpicking such a value would be inefficient and troublesome due to its dataset-specific nature. A practical approach is to determine the value with statistical information of the vector. In this work, we use the mean value for convenience. We empirically compare the performance using alternative thresholds in Appendix A.9. Let $D = \{i | s_i < \min(0, \frac{1}{F} \sum_{i=1}^{F} s_i)\}$ denote the set of unimportant components' indices. Finding the minimum between the mean value and 0 ensures the threshold is non-positive. This is to prevent components with positive scores from being sampled. The distortion vector $\mathbf{w}_{\text{dist}} = \frac{1}{\delta_s} \mathbb{1}_{\{i \in D\}} \odot |\mathbf{s}| \in \mathbb{R}_{\geq 0}^F$ modifies the unimportant frequency components to various extent. The scaling factor $\delta_s = \frac{1}{|D|} \sum_{i=1}^{F} \mathbb{1}_{\{i \in D\}} |s_i|$ controls the degree of distortion such that it is in accordance to each component's insignificance and no dramatic interference will be introduced. Because of the absolute value function, the least important frequency component gets amplified mostly in the distortion step. Lastly, we apply stop-gradient operation, i.e., $\mathbf{w}_{\text{dist}} = \text{stopgrad}(\mathbf{w}_{\text{dist}})$ because back-propagation is not desired for the distortion. Data-driven thresholding and scaling define the self-adaptive nature of modification on unimportant frequency components. By modifying these components, variance is infused into all timestamps.

**Overall Objective.** The Gumbel-Softmax reparameterization makes $\mathbf{w}_{\text{crit}}$ differentiable, which allows the joint training of automatic augmentation and the contrastive learning framework. Specifically, the contrastive model is optimized by pulling positive pairs together and pushing negative pairs apart through the InfoNCE loss (van den Oord et al., 2018), given by:

$$\mathcal{L}_{\text{CL}} = -\frac{1}{B} \sum_{x \in X} \log \frac{\exp(\text{sim}(h_x, \hat{h}_x)/\tau)}{\sum_{x' \in X} \exp(\text{sim}(h_x, \hat{h}_{x'})/\tau)}, \tag{5}$$

where $h_x = g_\phi(f_\theta(x))$, $\hat{h}_x = g_\phi(f_\theta(\mathcal{A}_\mathbf{s}(x)))$, $\text{sim}(\cdot, \cdot)$ denotes the similarity measurement implemented as the cosine similarity and $\tau$ is the temperature coefficient. Minimizing the InfoNCE loss is equivalent to maximizing the lower bound $\text{MI}_{\text{CL}}(\mathsf{x}, \mathcal{A}_\mathbf{s}(\mathsf{x}))$ of the mutual information $\text{MI}(\mathsf{x}, \mathcal{A}_\mathbf{s}(\mathsf{x}))$ (van den Oord et al., 2018), i.e., $\text{MI}(\mathsf{x}, \mathcal{A}_\mathbf{s}(\mathsf{x})) \leq \log(B) - \mathcal{L}_{\text{CL}} = \text{MI}_{\text{CL}}(\mathsf{x}, \mathcal{A}_\mathbf{s}(\mathsf{x}))$, where $B$ denotes the batch size. For $\mathcal{A}_\mathbf{s}(\cdot)$, directly applying InfoNCE results in a trivial solution of $\mathbf{s}$ that causes $\mathbf{w}_{\text{crit}}$ to become an all-one vector $\mathbf{1} \in \{1\}^F$, leaving the importance of frequency components ambiguous. This is because $x = \mathcal{F}^{-1}(\mathbf{1} \odot x_f)$. On the other hand, the critical components should keep and only keep the critical information, as the name suggests, i.e., $\text{MI}(\mathsf{x}_{\text{crit}}; \mathsf{x}) = \text{MI}(\mathsf{x}; \mathsf{y})$, where $\mathsf{x}_{\text{crit}} = \mathcal{F}^{-1}(\mathbf{w}_{\text{crit}} \odot x_f)$. Knowing that DFT is a reversible operation, we prove $\text{MI}(\mathsf{x}_{\text{crit}}; \mathsf{x}) = \text{MI}(\mathbf{w}_{\text{crit}} \odot x_f; x_f)$ and $\text{MI}(\mathsf{x}; \mathsf{y}) = \text{MI}(x_f; \mathsf{y})$ in the Appendix A.4. The orthogonal property of the Fourier basis reminds us that the frequency components are uncorrelated. In other words, $\text{MI}(\mathbf{w}_{\text{crit}} \odot x_f; x_f))$ keeps increasing as a higher proportion of frequency components

are identified as critical ones, as illustrated in Figure 4 in the Appendix, with proof provided in the Appendix A.4. The trivial solution falls on the right end of the red segment and the optimal proportion of critical components is pointed by the green arrow. To avoid the trivial solution and to achieve a good view, we regularize the proportion of critical components in complement to the InfoNCE loss. Specifically, we employ the L1-norm on $\mathbf{w}_{\text{crit}}$ as follows:

$$\mathcal{L}_{\text{reg}} = \frac{1}{F} \sum_{f=1}^{F} |w_{\text{crit}}^f|. \tag{6}$$

The regularization eliminates redundancy from identifying too many critical Fourier components, leading to compact selection and robust representation learning. The overall optimization problem is given by:

$$\mathbf{s}^*, \theta^*, \phi^* = \arg\min_{\mathbf{s}, \theta, \phi}(\mathcal{L}_{\text{CL}} + \lambda \cdot \mathcal{L}_{\text{reg}}), \tag{7}$$

where $\lambda$ is a hyper-parameter to balance the two losses. Note that there exists a unique value of critical component's proportion that makes $\text{MI}(\mathbf{x}_{\text{crit}}; \mathbf{x}) = \text{MI}(\mathbf{x}; \mathbf{y})$ happen, as shown in Figure 4. Meanwhile, as the hyper-parameter regularizes the proportion, $\lambda$ empirically exhibits stable performance over a range of values, as shown in the Appendix A.9.

**How Does the Learning Objective Benefit View Generation?**  Optimizing Eq. 7 is equivalent to maximize the lower bound for $\text{MI}(\mathbf{x}, \mathbf{x}_{\text{crit}})$ and minimize $\text{MI}(\mathcal{A}_{\mathbf{s}}(\mathbf{x}), \mathbf{x})$. The former occurs because optimizing $\mathbf{s}$ over the InfoNCE loss only maximizes the lower bound for $\text{MI}(\mathbf{x}, \mathbf{x}_{\text{crit}})$, due to the stop-gradient operation applied to the unimportant frequency component. The latter is achieved by the regularization term and the distortion applied to unimportant components. Combined with the Proposition 3 we have proved earlier, we prove the view generator trained on objective in Eq. 7 satisfies the optimality as defined in Definition 1. Moreover, unlike time-domain augmentations that disrupt the inter-correlations among timestamps and harm the semantics, FreRA independently modifies critical and unimportant components in the frequency domain, protecting the global semantics intact while introducing variance.

**Distinction to Existing Automatic Augmentation for Time Series Contrastive Learning**  At first glance, our method may seem similar to InfoTS (Luo et al., 2023) and AutoTCL (Zheng et al., 2024), but FreRA fundamentally differs in the view generation process, i.e., how it applies the reparameterization trick and where it disentangle the information. For detailed explanations, please refer to the Appendix A.5.

## 4 EXPERIMENTS

**Datasets**  To fully evaluate the model performance under different scenarios, we conduct extensive experiments on: (1) 3 large-scale datasets on HAR: UCIHAR (Anguita et al., 2012), MotionSense (MS) (Malekzadeh et al., 2019), and WISDM (Kwapisz et al., 2010); (2) the UEA archive (Bagnall et al., 2018): 30 multivariate time series datasets from various applications such as Human Activity Recognition (HAR), Motion classification, ECG classification, EEG/MEG classification, Audio Spectra Classification and so on; (3) the UCR archive (Dau et al., 2019): 100 univariate time series datasets collected from real-world scenarios; (4) a large-scale anomaly detection dataset: Fault Diagnosis (FD) (Lessmeier et al., 2016) aiming to detect and classify bearing damages from single-channel current signals of electric motors; (5) a large-scale HAR dataset for transfer learning scenario: SHAR (Micucci et al., 2016), which contains daily activity signals from 30 persons and is empirically observed to have large distribution gap among individuals (Qian et al., 2022).

**Baselines**  We compare FreRA against the following related baselines: (1) 11 commonly-used handcrafted time-domain (T) augmentations (Qian et al., 2022), including jitter, scaling, negation, permutation, shuffling, time-flipping, time-warping, resampling, rotation, permutation-and-jitter, jitter-and-scale; (2) 5 handcrafted frequency-domain (F) augmentations (Qian et al., 2022), including low-pass filter, high-pass filter, phase shift, amplitude and phase perturbation (fully), and amplitude and phase perturbation (partially); (3) 3 SOTA automatic augmentation for contrastive learning: InfoMin (Tian et al., 2020), InfoTS (Luo et al., 2023), and AutoTCL (Zheng et al., 2024); (4) 5 SOTA time series contrastive learning frameworks: TS2Vec (Yue et al., 2022b), TNC (Tonekaboni et al., 2021), TS-TCC (Eldele et al., 2023), TF-C (Zhang et al., 2022) and SoftCLT (Lee et al., 2024).

Table 1: The overall performance on all the datasets (unit: %). best(T) and best(F) record the highest performances among the selected sets of 11 time-domain augmentations and 5 frequency-domain augmentations. The best performance is highlighted in **bold**, and the second-best performance is underlined. $^*$ indicates FreRA significantly outperforms both best(T) and best(F) at the confidence level of 0.05 from paired t-test.

| Dataset | Metrics | FreRA (ours) | best(T) | best(F) | InfoMin$^+$ | InfoTS | AutoTCL | TS2Vec | TNC | TS-TCC | TF-C | SoftCLT |
|---|---|---|---|---|---|---|---|---|---|---|---|---|
| UCIHAR | ACC | **0.975**$^*$ | 0.959 | 0.960 | 0.967 | 0.967 | 0.697 | 0.959 | 0.568 | 0.924 | 0.875 | 0.961 |
| MS | ACC | **0.982**$^*$ | 0.956 | 0.970 | 0.971 | 0.967 | 0.691 | 0.945 | 0.526 | 0.915 | 0.811 | 0.962 |
| WISDM | ACC | **0.972**$^*$ | 0.942 | 0.950 | 0.959 | 0.915 | 0.760 | 0.939 | 0.543 | 0.889 | 0.839 | 0.952 |
| UEA Archive | Avg. ACC | **0.754**$^*$ | 0.684 | 0.686 | 0.693 | 0.714 | 0.742 | 0.704 | 0.670 | 0.668 | 0.298 | 0.751[1] |
|  | Avg. RANK | **2.133** | 5.967 | 5.800 | 5.500 | 3.967 | 2.600 | 4.967 | 6.433 | 6.033 | 9.276 | - |
| UCR Archive | Avg. ACC | **0.850**$^*$ | 0.723 | 0.744 | 0.718 | 0.849 | 0.598 | 0.845 | 0.776 | 0.780 | 0.542 | **0.850**[1] |
|  | Avg. RANK | 1.940 | 6.320 | 5.750 | 6.470 | **1.930** | 8.420 | 2.670 | 4.810 | 4.670 | 8.330 | - |

**Implementation Details** We use the predefined train-validation-test split if the dataset includes such information. Otherwise, we split each dataset with a ratio of 64%:16%:20%. For time-series classification datasets with class imbalance issues, we sample training instances with probabilities inversely proportional to their class sizes. We implement FreRA in PyTorch (Paszke et al., 2019) and conduct all experiments on an NVIDIA GeForce RTX 3090 GPU with 25 GB memory. Additional implementation details are included in Appendix A.6.

### 4.1 MAIN RESULTS ON TIME SERIES CLASSIFICATION TASKS

The overall results on all the datasets are presented in Table 1. Overall, FreRA consistently outperforms all the baselines on the three large HAR datasets and achieves the top average accuracy and ranking on both UEA and UCR archives. The detailed performances of the UEA and UCR archives are reported in Table 10 and Table 9 in the Appendix. FreRA achieves the best performance on 17 out of 30 datasets in the UEA archive. We credit the surprising performance to the frequency-refined views generated by FreRA. The empirical performance adequately illustrates that FreRA can effectively keep the semantic information from critical frequency components intact while infusing variance, boosting representation learning on all datasets. FreRA achieves leading performances not only on large-scale HAR datasets but also on extremely small datasets, e.g., AtrialFibrillation, DuckDuckGeese, and StandWalkJump within the UEA archive, whose training sets contain less than 100 samples, and the improvement over the second-best baselines is up to $8.7\%$ on average. This is not only credited to the effectiveness of FreRA in maintaining semantics but also to the lightweight and scalable design where the number of parameters is only half of the sequence length. It also indicates that FreRA provides robust performance across datasets of varying sizes. Although FreRA achieves an average ranking 0.01 lower than InfoTS on the UCR archive, when comparing FreRA to the baselines of the same backbone structure, i.e. best(T), best(F) and InfoTS$^+$, the improvement brought by the augmentation itself is significantly larger than the difference between InfoTS and TS2Vec. This indicates that FreRA offers stronger enhancement regardless of the backbone. We present a detailed analysis in the Appendix A.7. We also evaluate the performance of FreRA on the *anomaly detection task* using the Fault Diagnosis dataset and present the result and analysis in the Appendix A.8

### 4.2 EVALUATION ON TRANSFER LEARNING

Here, we evaluate the generalizability of the pre-trained encoder, which is crucial when there exists a misalignment between the per-training data and data from downstream tasks. The encoder is pre-trained on the source domains and adopted directly to an unseen target domain. Following (Qian et al., 2022), we conduct transfer learning in data-scarce and data-rich settings, where the number of source domains for training is 3 and 19 respectively. Table 2 records the results from the two settings. FreRA is shown to be more effective in learning generalizable encoders than all the baselines. This is because the views emphasizing the semantic-preserving patterns guide the training of the encoder and make it sensitive to the inherent global semantic information and robust to the unimportant information, i.e., distribution shift among different domains. Without effectively identifying critical

---

[1]The result is directly adopted from its original paper. As the results across all the datasets in the UEA and UCR archives are not provided, the ranking is not available.

Table 2: Classification performance in transfer learning setting on the SHAR dataset under different numbers of source domains. No. SD denotes the number of source domains for pre-training and TD denotes the index of the target domain. The best accuracy is highlighted in **bold**, and the second-best performance is underlined.

| No. SD | TD | FreRA (ours) | best(T) | best(F) | InfoMin$^+$ | InfoTS | AutoTCL | TS2Vec | TNC | TS-TCC | TF-C | SoftCLT |
|---|---|---|---|---|---|---|---|---|---|---|---|---|
| 3 | 1 | **0.602** | 0.599 | 0.495 | 0.537 | 0.367 | 0.464 | 0.430 | 0.133 | 0.495 | 0.349 | 0.505 |
| | 2 | **0.467** | 0.415 | 0.412 | 0.359 | 0.369 | 0.278 | 0.317 | 0.145 | 0.410 | 0.252 | 0.407 |
| | 3 | **0.665** | 0.582 | 0.599 | 0.516 | 0.516 | 0.414 | 0.523 | 0.217 | 0.464 | 0.568 | 0.530 |
| | 5 | **0.366** | 0.332 | 0.336 | 0.359 | 0.081 | 0.245 | 0.050 | 0.143 | 0.362 | 0.255 | 0.339 |
| 19 | 1 | **0.628** | 0.555 | 0.607 | 0.542 | 0.599 | 0.497 | 0.568 | 0.117 | 0.578 | 0.453 | 0.581 |
| | 2 | **0.652** | 0.583 | 0.571 | 0.563 | 0.455 | 0.372 | 0.640 | 0.148 | 0.647 | 0.456 | 0.581 |
| | 3 | **0.691** | 0.628 | 0.665 | 0.638 | 0.563 | 0.408 | 0.502 | 0.135 | 0.592 | 0.451 | 0.559 |
| | 5 | **0.698** | 0.617 | 0.638 | 0.601 | 0.638 | 0.430 | 0.658 | 0.204 | 0.612 | 0.466 | 0.567 |

Table 3: Effects of the two modification modules and the L1-norm regularization of FreRA. Results are averaged over 30 datasets from the UEA archive. The number in the bracket illustrates the accuracy gap with FreRA.

| | FreRA (ours) | w/o modification on critical components | w/o modification on noise components | w/o L1-norm regularization on $\mathbf{w}_{crit}$ |
|---|---|---|---|---|
| Avg. ACC | **0.754** | 0.690 (-0.064) | 0.695 (-0.059) | 0.690 (-0.064) |

and unimportant information, other SOTA baselines on automatic augmentation and time series contrastive learning fail to deliver promising performance in the transfer learning scenario.

### 4.3 ABLATION STUDIES

**Effect of Each Component.** In Table 3, we evaluate the effect of each component of FreRA, i.e. the identity modification on critical frequency components, the self-adaptive modification on unimportant frequency components, and the regularization term. To disallow the identity modification on critical components, we randomly sample a proportion of critical components instead of identifying their distribution in a data-driven way. The proportion is the same as $\mathcal{L}_{reg}$ from the last epoch of our approach to ensure fair comparison. To disallow the self-adaptive modification on unimportant frequency components, we set $\mathbf{w}_{dist}$ as an all-zero vector. To ignore the regularization term, we let hyper-parameter $\lambda$ be 0. Overall, removing any component deteriorates performance drastically. The semantic information and the distortion introduced are both crucial for downstream tasks. Between the two, the identity modification on critical components is slightly more important than the distortion on noisy components. It demonstrates the effectiveness of isolating critical and non-critical components from the frequency domain and applying the respective modifications accordingly. Last but not least, the L1-norm regularization is as crucial as the two frequency modification modules. The result demonstrates the importance of maintaining the inherent compact distribution of critical components.

**Sensitivity to Hyper-parameter** $\lambda$. Figure 5 in the Appendix shows the accuracy of FreRA on the 3 HAR datasets under varying $\lambda$ compared to their second-best baselines plotted in dashed lines for reference. The result demonstrates that the downstream task performance remains stable across different values of $\lambda$ and consistently better than the baseline, indicating FreRA is robust to the selection of the hyper-parameter's value. A detailed analysis is presented in the Appendix A.9.

More comprehensive ablation studies investigating the sensitivity to hyper-parameter $\lambda$, the performance of alternative contrastive learning frameworks, the effect of unimportant component selection mechanisms, and the robustness to Gaussian noise are presented in the Appendix A.9.

## 5 CONCLUSION

In this paper, we propose Frequency-Refined Augmentation (FreRA), a lightweight yet effective augmentation for time series contrastive learning on classification tasks. FreRA leverages the global, independent, and compact nature of the frequency domain to generate semantic-preserving views through independent modifications on separated frequency components. Its effectiveness is verified both theoretically and empirically. Experiments on 135 benchmark datasets from various applications demonstrate that FreRA is universally effective in contrastive learning and generalizes well in transfer learning scenarios. In addition, it is robust to hyper-parameter settings, flexible and effective when applied to various contrastive learning frameworks, and resilient to Gaussian noise added to the input.

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

# A APPENDIX

## A.1 DETAILS REGARDING FIGURE 1

Directly calculating the mutual information between the entire time series and the label is not trivial due to the curse of dimensionality. To address this, in Figure 1, we calculate the mutual information between each timestamp and the label. It visualizes the amount of semantic information preserved across all the timestamps. For instance, the value of the orange curve at timestamp $\forall t \in [1, ..., L]$ is $MI(\mathbf{x}^t; \mathbf{y})$ of the UCIHAR dataset, where $\mathbf{x}^t \in \mathbb{R}^D$ is the signal at timestamp $t$, $\mathbf{y}$ is the ground truth label. $L = 128$ and $D = 6$ are the length and dimension of the samples. Estimating the timestamp-wise mutual information, with a dimension of only 6, allows us to avoid the curse of dimensionality. In this plot, we do not intend to suggest a single timestamp alone is fully representative of the underlying semantics. Instead, the figure illustrates how the informative content varies across different augmentation functions. While a single timestamp may not directly indicate specific semantic meaning, the plot demonstrates the manipulation of the frequency domain benefits the augmented views. This is attributed to the undeteriorated critical components that are semantically informative.

## A.2 DISCUSSION ON PREDEFINED FREQUENCY-DOMAIN AUGMENTATIONS

Frequency-based predefined augmentations, such as high-pass and low-pass filters, require prior knowledge, such as the effective bandwidth of the dataset, to determine the selection of appropriate augmentation functions. Additionally, stochastic frequency-domain augmentations, such as phase-shift and augmentation in TF-C Zhang et al. (2022), introduce random noise that disrupts the critical information. As prior knowledge is not always accessible in the contrastive learning paradigm, and the compromised semantics caused by the frequency-domain augmentation have been observed from the brown line in Figure 1, predefined frequency-domain augmentations remain suboptimal for contrastive learning. Existing frequency-based augmentations do not fully leverage the advantages of the frequency domain. To this end, we reanalyze the benefits of the frequency domain and deliberately design a frequency-based augmentation to address the aforementioned issues and fully utilize its advantages.

## A.3 PROPERTIES OF THE DISCRETE FOURIER TRANSFORM (DFT)

### A.3.1 CONJUGATE SYMMETRIC.

Given a signal $x(n) \in \mathbb{R}, n \in \{0, 1, ..., L-1\}$, its DFT $X(m) \in \mathbb{C}, m \in \{0, 1, ..., L-1\}$ is conjugate symmetric, i.e., $X(L-m) = \overline{X(m)}$. The proof is as follows:

$$
\begin{aligned}
X(L-m) &= \sum_{n=0}^{L-1} x(n) e^{-\frac{2\pi i}{L}(L-m)n} \\
&= \sum_{n=0}^{L-1} x(n) e^{\frac{2\pi i}{L}mn} \\
&= \overline{X(m)}.
\end{aligned}
\tag{8}
$$

Converting back to our notation, $X(m) = \overline{X(L-m)} = x_f^{L-m+1}$, which explains the second condition of $X(m)$ in Eq. 3. The Conjugate Symmetry allows only half of the DFT signal to recover the entire time series, which also justifies why FreRA manipulates only half of the frequency components, i.e., $F = \lfloor L/2 \rfloor + 1$. This property allows FreRA to have a lightweight structure.

### A.3.2 ORTHOGONAL OF FOURIER BASIS.

The inverse DFT, $x(n) = \frac{1}{L} \sum_{m=0}^{L-1} X(m) e^{\frac{2\pi i}{L}mn}$ decompose the time domain on the Fourier basis $\mathbf{u}_m = [e^{\frac{2\pi i}{L}mn} | n = 0, 1, ..., L-1]^T \in \mathbb{C}^L$, where frequency components $X(m)$ are the coefficients with respect to the Fourier basis. The orthogonal property of Fourier basis, i.e., $\langle \mathbf{u}_m, \mathbf{u}_q \rangle = 0$ if $m \neq q$, is proved below.

$$\langle \mathbf{u}_m, \mathbf{u}_q \rangle = \mathbf{u}_m^T \overline{\mathbf{u}_q}$$

$$= \sum_{n=0}^{L-1} e^{\frac{2\pi i}{L} mn} e^{-\frac{2\pi i}{L} qn}$$

$$= \sum_{n=0}^{L-1} e^{\frac{2\pi i}{L}(m-q)n}$$

$$\text{(the sum of a geometric series follows: } \sum_{n=0}^{L-1} r^n = \frac{1-r^L}{1-r})$$

$$= \frac{1 - e^{\frac{2\pi i}{L}L(m-q)}}{1 - e^{\frac{2\pi i}{L}(m-q)}}$$

$$(e^{\frac{2\pi i}{L}L(m-q)} = 1 \text{ and } e^{\frac{2\pi i}{L}(m-q)} \neq 1 \text{ if } m \neq q)$$

$$= 0$$

(9)

### A.4 Proofs of Propositions

**Proposition 1.** *(Conservation of Entropy) Let $x$ and $x_f$ be the random variables denoting the time series in the time domain and the frequency domain respectively, then we have $H(x) = H(x_f)$.*

*Proof.* Since the DFT is a one-to-one invertible transformation, we have $p(x) = p(x_f)$.

$$\begin{aligned} \mathrm{H}(x) &= \sum_x p(x) \log p(x) \\ &= \sum_{x_f} p(x_f) \log p(x_f) \\ &= \mathrm{H}(x_f) \end{aligned}$$

(10)

$\square$

**Proposition 2.** *(Conservation of Mutual Information) Let $x$, $x_f$, and $y$ be the random variables denoting the time series in the time domain and the frequency domain, and their corresponding label respectively, then we have $MI(x; y) = MI(x_f; y)$.*

*Proof.* Since the DFT does not alter the label of the time series variable, we have $p(x, y) = p(x_f, y)$.

$$\begin{aligned} \mathrm{MI}(x; y) &= \mathrm{H}(y) - \mathrm{H}(y|x) \\ &= \mathrm{H}(y) - \sum_{x,y} p(x,y) \log \frac{p(x,y)}{p(x)} \\ &= \mathrm{H}(y) - \sum_{x_f,y} p(x_f,y) \log \frac{p(x_f,y)}{p(x_f)} \\ &= \mathrm{MI}(x_f; y) \end{aligned}$$

(11)

Similarly, we can proof $\mathrm{MI}(x; \tilde{x}) = \mathrm{MI}(x_f; \tilde{x}_f)$, where random variable $x$ and $\tilde{x}$ denotes two time series and $x_f$ and $\tilde{x}_f$ denotes their frequency-domain counterpart. $\square$

**Proposition 3.** *With the reliable assumption that the noisy frequency components are independent to the label, FreRA is a semantic preserving transformation, i.e., $MI(\mathcal{A}_\mathbf{s}(x); y) = MI(x; y)$.*

*Proof.* Let $x_f^{\mathrm{crit}} = \mathbf{w}_{\mathrm{crit}} \odot x_f$ and $x_f^{\mathrm{dist}} = (\mathbf{1} - \mathbf{w}_{\mathrm{crit}}) \odot x_f$ denote the critical and noisy frequency components respectively. Knowing $x_f^{\mathrm{crit}}$ and $x_f^{\mathrm{dist}}$ are independent, we have

$$\mathrm{H}(x_f) = \mathrm{H}(x_f^{\mathrm{crit}}) + \mathrm{H}(x_f^{\mathrm{dist}}).$$

(12)

Then we show that

$$
\begin{aligned}
\mathrm{MI}(\mathsf{x}_f; \mathsf{y}) &= \mathrm{H}(\mathsf{x}_f) - \mathrm{H}(\mathsf{x}_f; \mathsf{y}) \\
&= \mathrm{H}(\mathsf{x}_f^{\mathrm{crit}}) + \mathrm{H}(\mathsf{x}_f^{\mathrm{dist}}) - \mathrm{H}(\mathsf{x}_f^{\mathrm{crit}}, \mathsf{x}_f^{\mathrm{dist}}|\mathsf{y}) \\
&\quad (\mathsf{x}_f^{\mathrm{dist}}, \text{ as irrelevant components, is independent to } \mathsf{y}) \\
&= \mathrm{H}(\mathsf{x}_f^{\mathrm{crit}}) + \mathrm{H}(\mathsf{x}_f^{\mathrm{dist}}) - (\mathrm{H}(\mathsf{x}_f^{\mathrm{crit}}|\mathsf{y}) + \mathrm{H}(\mathsf{x}_f^{\mathrm{dist}})) \\
&= \mathrm{H}(\mathsf{x}_f^{\mathrm{crit}}) - \mathrm{H}(\mathsf{x}_f^{\mathrm{crit}}|\mathsf{y}) \\
&= \mathrm{MI}(\mathsf{x}_f^{\mathrm{crit}}; \mathsf{y}).
\end{aligned}
\tag{13}
$$

Similarly,

$$
\begin{aligned}
\mathrm{MI}(\mathcal{A}_{\mathbf{s}}(\mathsf{x}); \mathsf{y}) &= \mathrm{MI}((\mathbf{w}_{\mathrm{crit}} + \mathbf{w}_{\mathrm{dist}}) \odot \mathsf{x}_f; \mathsf{y}) \\
&= \mathrm{H}((\mathbf{w}_{\mathrm{crit}} + \mathbf{w}_{\mathrm{dist}}) \odot \mathsf{x}_f) - \mathrm{H}((\mathbf{w}_{\mathrm{crit}} + \mathbf{w}_{\mathrm{dist}}) \odot \mathsf{x}_f; \mathsf{y}) \\
&= \mathrm{H}(\mathbf{w}_{\mathrm{crit}} \odot \mathsf{x}_f) + \mathrm{H}(\mathbf{w}_{\mathrm{dist}} \odot \mathsf{x}_f) - \mathrm{H}((\mathbf{w}_{\mathrm{crit}} \odot \mathsf{x}_f + \mathbf{w}_{\mathrm{dist}} \odot \mathsf{x}_f|\mathsf{y}) \\
&\quad (\mathbf{w}_{\mathrm{dist}} \odot \mathsf{x}_f \text{ is independent to } \mathsf{y}) \\
&= \mathrm{H}(\mathsf{x}_f^{\mathrm{crit}}) + \mathrm{H}(\mathbf{w}_{\mathrm{dist}} \odot \mathsf{x}_f) - (\mathrm{H}(\mathsf{x}_f^{\mathrm{crit}}|\mathsf{y}) + \mathrm{H}(\mathbf{w}_{\mathrm{dist}} \odot \mathsf{x}_f)) \\
&= \mathrm{H}(\mathsf{x}_f^{\mathrm{crit}}) - \mathrm{H}(\mathsf{x}_f^{\mathrm{crit}}|\mathsf{y}) \\
&= \mathrm{MI}(\mathsf{x}_f^{\mathrm{crit}}; \mathsf{y}).
\end{aligned}
\tag{14}
$$

Applying Proposition. 2, we have $\mathrm{MI}(\mathcal{A}_{\mathbf{s}}(\mathsf{x}); \mathsf{y}) = \mathrm{MI}(\mathsf{x}; \mathsf{y})$. □

**Proposition 4.** *$MI(\mathbf{w}_{crit}) \odot \mathsf{x}; \mathsf{x})$ is monotonically increasing w.r.t the proportion of critical components.*

*Proof.*

$$
\begin{aligned}
\mathrm{MI}(\mathbf{w}_{\mathrm{crit}} \odot \mathsf{x}_f; \mathsf{x}_f) &= \mathrm{H}(\mathsf{x}_f) - H(\mathsf{x}_f|\mathbf{w}_{\mathrm{crit}} \odot \mathsf{x}_f) \\
&= \mathrm{H}(\mathsf{x}_f) - H(\mathbf{w}_{\mathrm{crit}} \odot \mathsf{x}_f, (\mathbf{1} - \mathbf{w}_{\mathrm{crit}}) \odot \mathsf{x}_f|\mathbf{w}_{\mathrm{crit}} \odot \mathsf{x}_f) \\
&\quad (\mathbf{w}_{\mathrm{crit}} \odot \mathsf{x}_f \text{ and } (\mathbf{1} - \mathbf{w}_{\mathrm{crit}}) \odot \mathsf{x}_f \text{ are independent since they lie on the orthogonal basis}) \\
&= \mathrm{H}(\mathsf{x}_f) - H((\mathbf{1} - \mathbf{w}_{\mathrm{crit}}) \odot \mathsf{x}_f) \\
&= \mathrm{H}(\mathsf{x}_f) - \sum_{i=1}^{F} \mathbb{1}_{\{1-\mathbf{w}_{\mathrm{crit}}^i=1\}} \mathrm{H}(\mathsf{x}_f^i)
\end{aligned}
\tag{15}
$$

Since the first term $\mathrm{H}(\mathsf{x}_f)$ is fixed, and the second term $\sum_{i=1}^{F} \mathbb{1}_{\{1-\mathbf{w}_{\mathrm{crit}}^i=1\}} \mathrm{H}(\mathsf{x}_f^i)$ decreases as the proportion of critical components increases, we prove the monotonic increasing of $\mathrm{MI}(\mathbf{w}_{\mathrm{crit}} \odot \mathsf{x}_f; \mathsf{x}_f)$ w.r.t the proportion of critical components. As the proportion becomes 1, i.e., all the frequency components are identified as critical ones, $\mathrm{MI}(\mathbf{w}_{\mathrm{crit}} \odot \mathsf{x}_f; \mathsf{x}_f) = \mathrm{H}(\mathsf{x}_f)$, as we plot in Figure 4. □

## A.5 DISTINCTION TO EXISTING AUTOMATIC AUGMENTATION FOR TIME SERIES CONTRASTIVE LEARNING

At first glance, our method may seem to resemble InfoTS (Luo et al., 2023), since it also leverages the same reparameterization trick to facilitate the view generation. However, their $p_i$ indicates the probability of sampling a predefined transformation $\mathcal{T}_i(\cdot)$, i.e., $\mathcal{A}_{\mathrm{InfoTS}}(x) = \frac{1}{m} \sum_{i=1}^{m} \mathrm{Gumbel\text{-}Softmax}(p_i)\mathcal{T}_i(x)$. It fails to handle the noise and artifacts introduced by predefined augmentations $\mathcal{T}_i(\cdot)$. On the contrary, our approach elegantly eliminates the dependency on $\mathcal{T}_i(\cdot)$ by preserving critical elements and modifying the noise elements in the frequency domain. This more effectively enables preserving contextual-related information in the generated views while infusing variance. FreRA also appears similar to AutoTCL (Zheng et al., 2024) in the sense that it disentangles the informative information of the time series from the noisy ones. However, performing the disentanglement on the time domain disrupts the periodicity and inter-dependencies among timestamps in the real world and hinders the semantics from the input space. Conversely, we disentangle the information in the frequency domain and leverage its advantages over the time domain: global, independent, and compact, to better facilitate the view generation.

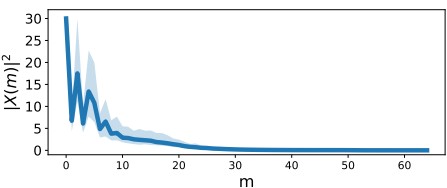
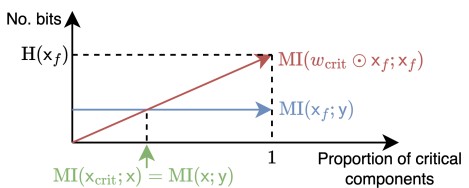

Figure 3: Take the UCIHAR dataset as an example, the energy in the frequency domain $E = \frac{1}{L}\sum_{m=0}^{L-1}|X(m)|^2$ is mostly concentrated in a compact set of frequency components, whose frequency are the ten lowest. The solid line represents the average energy for the frequency components in the UCIHAR dataset, and the shaded area indicates the range.

Figure 4: We aim to achieve the intersection point pointed by the green arrow where $\mathrm{MI}(\mathsf{x}_{\mathrm{crit}};\mathsf{x}) = \mathrm{MI}(\mathsf{x};\mathsf{y})$, meaning the critical frequency components keep and only keep the semantic information. The linearity of $\mathrm{MI}(\mathbf{w}_{\mathrm{crit}}\odot\mathsf{x}_f;\mathsf{x}_f)$ is for illustration purposes only.

### A.6 IMPLEMENTATION DETAILS

For the predefined time-domain and frequency-domain augmentations, we follow the parameter settings from (Qian et al., 2022). For the InfoMin baseline, we apply its adversarial objective to replace our regularization term. To make it suitable for time series, we use our frequency-domain refinement to substitute the flow-based view generator which is designed for images. This implementation makes it benefit from our frequency-enhanced approach and we denote this baseline as InfoMin$^+$. For other baselines, we adopt the results from (Zheng et al., 2024; Lee et al., 2024) if they are available. Otherwise, we use the publicly available implementation and fine-tune the model as suggested in the original papers.

Fully-convolutional Network (FCN) (Wang et al., 2017) with an output dimension 128 is adopted as the encoder $f_\theta$. The batch size is selected from $\{256, 128, 64, 32, 16, 5\}$ according to the scale of the dataset, and the maximum training epoch is set to 200 for all the experiments. The learning rate is selected from $\{0.03, 0.01, 0.003, 0.001\}$. We adopt SGD optimizer to train the contrastive model and Adam optimizer for $\mathbf{s}$. For the hyper-parameter setting, we select discretization temperature $\tau_w$ from $\{0.1, 0.2\}$, and fix temperature coefficient $\tau$ to be 0.2. $\lambda$ is searched from $\{0.1, 0.3, 1, 3, 10, 30\}$. The projector $g_\phi$ is a two-layer MLP, with hidden and output dimensions 128.

To evaluate the performance, we employ the commonly used linear evaluation protocol. We first jointly train FreRA and the contrastive learning model, then we discard other components and keep only the pre-trained encoder $f_{\theta*}$ frozen and train a linear classifier on top of it, as illustrated in the lower right corner of Figure 2. For time series classification tasks, we record the best accuracy (ACC) as the evaluation metric. For anomaly detection tasks, we record both the best accuracy and the Macro-F1 score.

### A.7 ADDITIONAL ANALYSIS FOR THE MAIN RESULT ON TIME SERIES CLASSIFICATION TASKS

From Table 1, Table 9, and Table 10, we conclude that frequency-domain augmentations outrun time-domain augmentations in general. This endorses our motivation that the frequency perspective is superior to its time-domain counterpart in preserving global semantics. InfoMin$^+$ exceeds other baselines on the three large HAR datasets, which demonstrates the efficacy of its objective. However, the performance gap between InfoMin$^+$ and ours indicates that directly applying an adversarial objective in the frequency domain is not customized for our approach and causes conflict in representation learning. On the other hand, our specially designed objective better suits the frequency-domain refinement. The 5 SOTA time-series contrastive learning frameworks with carefully designed architectures and objectives become uncompetitive compared to FreRA.

It is worth noting that our datasets cover multiple applications, diverse data scales, and various types of sensor modalities. Notably, FreRA receives the best overall performance on them, which proves that our approach provides a unified view generation approach and can be flexibly applied to various time-series applications.

Table 4: Performance on anomaly detection task on the Fault Diagnosis dataset. Each row corresponds to a setting where the pre-training set includes domains {a, bd, c} \ Target Domain, and the Target Domain is used for evaluation. The best accuracy is highlighted in **bold**, and the second-best performance is underlined.

| Target Domain | Metrics | FreRA (ours) | best(T) | best(F) | InfoMin$^+$ | InfoTS | AutoTCL | TS2Vec | TNC | TS-TCC | TF-C | SoftCLT |
|---|---|---|---|---|---|---|---|---|---|---|---|---|
| a | ACC | **0.620** | 0.574 | 0.519 | 0.613 | 0.461 | 0.496 | 0.468 | 0.440 | 0.296 | 0.455 | 0.608 |
| | Macro-F1 | **0.671** | 0.638 | 0.508 | 0.644 | 0.485 | 0.484 | 0.468 | 0.302 | 0.353 | 0.208 | 0.639 |
| bd | ACC | **0.859** | 0.826 | 0.767 | 0.807 | 0.731 | 0.433 | 0.802 | 0.455 | 0.823 | 0.455 | 0.808 |
| | Macro-F1 | **0.895** | 0.856 | 0.817 | 0.853 | 0.798 | 0.471 | 0.848 | 0.300 | 0.755 | 0.208 | 0.853 |
| c | ACC | **0.819** | 0.810 | 0.736 | 0.812 | 0.742 | 0.482 | 0.677 | 0.465 | 0.557 | 0.455 | 0.775 |
| | Macro-F1 | **0.858** | 0.794 | 0.755 | 0.848 | 0.781 | 0.456 | 0.747 | 0.314 | 0.617 | 0.208 | 0.825 |

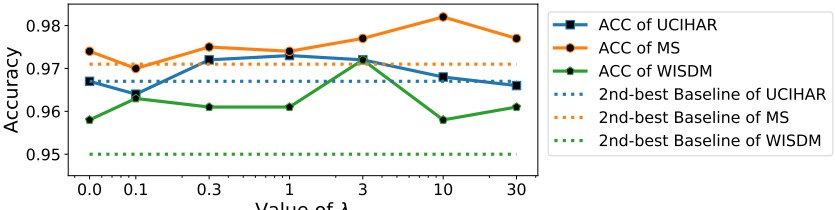

Figure 5: Performance of FreRA on the 3 HAR datasets under varying $\lambda$, in comparison to their second-best baselines.

Table 5: The performance of the three large HAR datasets on alternative time series contrastive learning models.

| Dataset | TS2Vec (InfoNCE) | | TS-TCC (InfoNCE) | | SoftCLT (InfoNCE) | |
|---|---|---|---|---|---|---|
| | FreRA (ours) | original augmentation | FreRA (ours) | original augmentation | FreRA (ours) | original augmentation |
| UCIHAR | **0.970** | 0.959 | **0.944** | 0.924 | **0.969** | 0.961 |
| MS | **0.968** | 0.945 | **0.959** | 0.915 | **0.974** | 0.962 |
| WISDM | **0.957** | 0.939 | **0.962** | 0.889 | **0.956** | 0.952 |

Table 6: The performance of the three large HAR datasets on alternative contrastive learning models originally designed for the vision domain.

| Dataset | SimCLR (InfoNCE) | | | SimCLR (NT-Xent) | | | BYOL (Cosine Similarity) | | |
|---|---|---|---|---|---|---|---|---|---|
| | FreRA (ours) | best(T) | best(F) | FreRA (ours) | best(T) | best(F) | FreRA (ours) | best(T) | best(F) |
| UCIHAR | **0.975** | 0.959 | 0.960 | **0.972** | 0.951 | 0.955 | **0.960** | 0.940 | 0.937 |
| MS | **0.982** | 0.956 | 0.970 | **0.979** | 0.969 | 0.965 | **0.983** | 0.968 | 0.954 |
| WISDM | **0.972** | 0.942 | 0.950 | **0.966** | 0.941 | 0.952 | **0.952** | 0.942 | 0.928 |

## A.8 Evaluation on Anomaly Detection Tasks

We evaluate the performance of FreRA on the anomaly detection task using the Fault Diagnosis dataset and present the results in Table 4. The signals are collected under 4 different operation settings {a, b, c, d}. Observing the negligible domain gap between signals from settings 'b' and 'd', we randomly sample half of the data from each setting and combine them as a new domain 'bd'. Considering the highly imbalanced class distribution, we include the Macro-F1 score as another evaluation metric. FreRA outperforms all the baselines on both evaluation metrics, which demonstrates its strong performance in applications beyond classification.

## A.9 Ablation Studies

**Sensitivity to Hyper-parameter $\lambda$.** In Figure 5, UCIHAR, MS and WISDM achieve peak performances at $\lambda = 1, 10, 3$ respectively. On the left of the peak, the performance is suboptimal because redundant frequency components are included in the critical components. On the right of the peaks,

Table 7: The performance of applying different statistical information to select unimportant components.

| Dataset | mean (ours) | median | mean+std |
|---------|-------------|--------|----------|
| UICHAR | 0.975 | 0.971 | 0.972 |
| MS | 0.982 | 0.976 | 0.975 |
| WISDM | 0.972 | 0.963 | 0.963 |

Table 8: Performance comparison of FreRA on different datasets with and without Gaussian noise

| Dataset | w Gaussian noise | w/o Gaussian noise |
|---------|------------------|--------------------|
| UCIHAR | 0.970 | 0.975 |
| MS | 0.975 | 0.982 |
| WISDM | 0.964 | 0.972 |

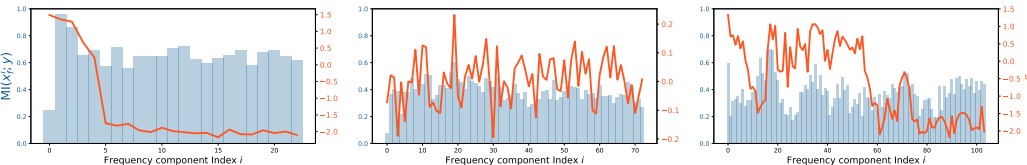

Figure 6: Despite the diverse distributions of global semantics across three datasets (Libras, ArticularyWor-dRecognition, and Epilepsy), as shown in the blue-grey bar plots, the learned vector **s**, represented by the orange lines, consistently captures the inherent critical information by assigning higher values to the most semantically relevant frequency components (those of high values in the bar plots).

incomplete critical information leads to degraded performances. The peak values indicate that FreRA learns to preserve only critical frequency components and distort the irrelevant components, achieving the optimal view for representation learning.

**On Alternative Contrastive Learning Frameworks.** Due to its meticulous design, FreRA can be seamlessly integrated with different contrastive models in a plug-and-play manner. In Table 5 and Table 6, we apply FreRA to five alternative contrastive learning models: (1) three time-series contrastive models TS2Vec (Yue et al., 2022a), TS-TCC (Eldele et al., 2023), and SoftCLT (Lee et al., 2024) with their default augmentations as baselines and (2) two general purpose contrastive learning models originally designed for the vision domain, BYOL (Grill et al., 2020) and SimCLR (Chen et al., 2020) with the best time-domain and frequency-domain augmentations as baselines. For SimCLR, despite the NT-Xent loss originally applied in SimCLR, we also use InfoNCE as the loss function, which forms the framework we use in our main result. The same usage has been deployed in Yeh et al. (2022) and Wu et al. (2024) as well. Our current evaluation covers 5 contrastive learning frameworks and 3 types of contrastive loss functions. All the models differ in network design and optimization objectives. It is worth noting that the contrastive losses used in TS-TCC, TS2Vec, and SoftCLT are different variants of InfoNCE, each with its unique formulation. The results presented consistently demonstrate that FreRA is a plug-and-play method that effectively enhances existing contrastive learning frameworks. This experiment highlights the flexibility and adaptability of our approach.

**Effect of Unimportant Component Selection Mechanisms.** To evaluate how the choice of statistical measurement in $D$ affects the final results, we conduct an ablation study comparing the performances when using mean, median, and mean+std of vector **s** as the threshold. The results are shown in Table 7. All the choices outperform the baseline performances in Table 1 and the mean value achieves the best performance among them.

**Robustness to Gaussian Noise.** In Table 8, we present the performance of FreRA on the three HAR datasets in the presence of Gaussian noise. The Gaussian noise has a mean of 0 and a standard deviation of 0.8. Despite a slight degradation in performance, FreRA still outperforms all the baselines shown in Table 1 when there is Gaussian noise in the input time series. Note that Gaussian noise is absent in all the baselines. It demonstrates the robust performance of FreRA with respect to the Gaussian noise in the time series.

**Vector s Captures the Inherent Semantic Distribution in the Frequancy Domain.** To verify the effectiveness of FreRA, in Figure 6, we visualize the learned parameter vector **s**, as compared to the ground truth semantics distribution in the frequency domain, on three datasets, including Libras, ArticularyWordRecognition, and Epilepsy. Specifically, we use the mutual information (MI) between the frequency components with the label to quantify the ground truth importance of frequency components and presented by blue-grey bar plots. The distribution of important frequency components varies across datasets. The important components are distributed in low frequencies, middle frequencies, and across multiple frequencies in these datasets, respectively. The learned vector **s** which determines the importance scores of all the frequency components is presented with the orange line plots. Despite diverse distributions, **s** consistently captures the inherent critical information by learning to assign higher values to the most semantically relevant frequency components.

### A.10   ADDITIONAL RESULTS

Full results of multivariate time series classification on the UEA archive and univariate time series classification on the UCR archive are presented in Table 10 and Table 9. The full result of the commonly used sets of 11 time-domain augmentations and 5 frequency-domain augmentations on the 3 HAR datasets are shown in Table 11 and Table 12 respectively.

Table 9: The overall classification result of 100 univariate time series datasets from the UCR archive. The best performance is highlighted in **bold**.

| Dataset | FreRA (Ours) | best(T) | best(F) | InfoMin | InfoTS | AutoTCL | TS2Vec | TNC | TS-TCC | TF-C |
|---|---|---|---|---|---|---|---|---|---|---|
| ACSF1 | 0.760 | 0.660 | 0.470 | 0.580 | 0.850 | 0.480 | **0.910** | 0.730 | 0.730 | 0.100 |
| AllGestureWiimoteX | 0.707 | 0.526 | 0.561 | 0.549 | 0.630 | 0.517 | **0.777** | 0.703 | 0.697 | 0.100 |
| AllGestureWiimoteY | 0.746 | 0.620 | 0.601 | 0.611 | 0.686 | 0.624 | **0.793** | 0.699 | 0.741 | 0.100 |
| AllGestureWiimoteZ | 0.707 | 0.581 | 0.573 | 0.577 | 0.629 | 0.576 | **0.770** | 0.646 | 0.689 | 0.100 |
| BeetleFly | **1.000** | 0.700 | 0.900 | 0.800 | 0.950 | 0.650 | 0.900 | 0.850 | 0.800 | 0.450 |
| BirdChicken | **1.000** | 0.750 | 0.850 | 0.800 | 0.900 | 0.550 | 0.800 | 0.750 | 0.650 | 0.500 |
| BME | **1.000** | 0.920 | 0.940 | 0.967 | **1.000** | 0.640 | 0.993 | 0.973 | 0.933 | 0.630 |
| CBF | **1.000** | 0.997 | 0.957 | 0.967 | 0.999 | 0.707 | **1.000** | 0.983 | 0.998 | 0.686 |
| Chinatown | **0.988** | 0.910 | 0.886 | 0.939 | **0.988** | 0.983 | 0.968 | 0.977 | 0.983 | 0.904 |
| CinCECGTorso | **0.968** | 0.912 | **0.968** | 0.914 | 0.928 | 0.305 | 0.827 | 0.669 | 0.671 | 0.248 |
| Coffee | **1.000** | 0.964 | 0.964 | **1.000** | **1.000** | 0.750 | **1.000** | **1.000** | **1.000** | 0.464 |
| Computers | **0.776** | 0.684 | 0.700 | 0.676 | 0.748 | 0.468 | 0.660 | 0.684 | 0.704 | 0.644 |
| Crop | 0.755 | 0.569 | 0.566 | 0.561 | **0.766** | 0.608 | 0.756 | 0.738 | 0.742 | 0.632 |
| DiatomSizeReduction | 0.980 | 0.817 | 0.948 | 0.905 | **0.997** | 0.676 | 0.987 | 0.993 | 0.977 | 0.301 |
| DistalPhalanxOutlineAgeGroup | 0.799 | 0.655 | 0.755 | 0.645 | 0.763 | 0.640 | 0.727 | 0.741 | 0.755 | 0.732 |
| DistalPhalanxOutlineCorrect | **0.804** | 0.627 | 0.670 | 0.734 | 0.801 | 0.583 | 0.775 | 0.754 | 0.754 | 0.683 |
| DistalPhalanxTW | **0.755** | 0.676 | 0.719 | 0.669 | 0.727 | 0.597 | 0.698 | 0.669 | 0.676 | 0.669 |
| DodgerLoopDay | 0.600 | 0.275 | 0.325 | 0.388 | **0.675** | 0.338 | 0.562 | - | - | 0.150 |
| DodgerLoopGame | 0.942 | 0.833 | 0.797 | 0.855 | 0.942 | 0.725 | 0.841 | - | - | 0.522 |
| DodgerLoopWeeken | **0.993** | 0.935 | 0.949 | 0.978 | 0.986 | 0.920 | 0.964 | - | - | 0.739 |
| Earthquakes | 0.820 | 0.748 | 0.748 | 0.748 | **0.821** | 0.748 | 0.748 | 0.748 | 0.748 | 0.748 |
| ECG200 | 0.890 | 0.830 | 0.840 | 0.800 | 0.930 | 0.700 | 0.920 | 0.830 | 0.880 | **0.940** |
| ECG5000 | **0.948** | 0.935 | 0.940 | 0.926 | 0.945 | 0.900 | 0.935 | 0.937 | 0.941 | 0.938 |
| ECGFiveDays | **1.000** | 0.998 | 0.987 | 0.990 | **1.000** | 0.821 | **1.000** | 0.999 | 0.878 | 0.972 |
| ElectricDevices | 0.657 | 0.609 | 0.599 | 0.609 | 0.702 | 0.562 | **0.721** | 0.700 | 0.686 | 0.560 |
| EOGHorizontalSignal | **0.597** | 0.434 | 0.508 | 0.470 | 0.572 | 0.293 | 0.544 | 0.442 | 0.401 | 0.083 |
| EOGVerticalSignal | 0.489 | 0.320 | 0.423 | 0.312 | 0.459 | 0.290 | **0.503** | 0.392 | 0.376 | 0.144 |
| FaceAll | 0.888 | 0.628 | 0.728 | 0.658 | **0.929** | 0.689 | 0.805 | 0.766 | 0.813 | 0.714 |
| FaceFour | 0.864 | 0.773 | 0.773 | 0.773 | 0.818 | 0.205 | **0.932** | 0.659 | 0.773 | 0.330 |
| FacesUCR | 0.866 | 0.861 | 0.794 | 0.760 | 0.913 | 0.544 | **0.930** | 0.789 | 0.863 | 0.779 |
| FordA | 0.943 | 0.905 | 0.902 | 0.917 | 0.915 | 0.494 | **0.948** | 0.902 | 0.930 | 0.537 |
| FordB | **0.832** | 0.775 | 0.794 | 0.780 | 0.785 | 0.493 | 0.807 | 0.733 | 0.815 | 0.474 |
| FreezerRegularTrain | 0.994 | 0.804 | 0.856 | 0.820 | **0.996** | 0.717 | 0.986 | 0.991 | 0.989 | 0.742 |
| FreezerSmallTrain | **0.988** | 0.787 | 0.735 | 0.811 | **0.988** | 0.721 | 0.894 | 0.982 | 0.979 | 0.501 |
| Fungi | 0.941 | 0.667 | 0.667 | 0.677 | 0.946 | 0.263 | **0.962** | 0.527 | 0.753 | 0.860 |
| GestureMidAirD1 | **0.638** | 0.315 | 0.431 | 0.308 | 0.592 | 0.423 | 0.631 | 0.431 | 0.369 | 0.038 |
| GestureMidAirD2 | **0.608** | 0.292 | 0.138 | 0.269 | 0.492 | 0.177 | 0.515 | 0.362 | 0.254 | 0.038 |
| GesturePebbleZ1 | 0.779 | 0.581 | 0.610 | 0.506 | 0.802 | 0.650 | **0.930** | 0.378 | 0.395 | 0.163 |
| GesturePebbleZ2 | 0.722 | 0.614 | 0.551 | 0.475 | 0.842 | 0.424 | **0.873** | 0.316 | 0.430 | 0.152 |
| GunPoint | **1.000** | 0.887 | 0.993 | 0.933 | **1.000** | 0.800 | 0.987 | 0.967 | 0.993 | 0.573 |
| GunPointAgeSpan | 0.984 | 0.921 | 0.908 | 0.908 | **1.000** | 0.639 | 0.994 | 0.984 | 0.994 | 0.927 |
| GunPointMaleVersusFemale | **1.000** | 0.835 | 0.839 | 0.832 | **1.000** | 0.718 | **1.000** | 0.994 | 0.997 | 0.987 |
| GunPointOldVersusYoung | **1.000** | **1.000** | **1.000** | **1.000** | **1.000** | 0.981 | **1.000** | **1.000** | **1.000** | **1.000** |
| Ham | 0.810 | 0.790 | 0.705 | 0.686 | **0.838** | 0.533 | 0.724 | 0.752 | 0.743 | 0.752 |
| HandOutlines | 0.900 | 0.876 | 0.886 | 0.881 | **0.946** | 0.662 | 0.930 | 0.930 | 0.724 | 0.641 |
| Haptics | 0.487 | 0.471 | 0.487 | 0.406 | **0.546** | 0.334 | 0.536 | 0.474 | 0.396 | 0.208 |
| Herring | **0.703** | 0.594 | 0.609 | 0.594 | 0.656 | 0.594 | 0.641 | 0.594 | 0.594 | 0.594 |
| HouseTwenty | **0.983** | 0.891 | 0.706 | 0.681 | 0.924 | 0.655 | 0.941 | 0.782 | 0.790 | 0.571 |
| InlineSkate | 0.353 | 0.258 | 0.318 | 0.242 | **0.424** | 0.193 | 0.415 | 0.378 | 0.347 | 0.155 |
| InsectEPGRegularTrain | **1.000** | **1.000** | **1.000** | **1.000** | **1.000** | **1.000** | **1.000** | **1.000** | **1.000** | **1.000** |
| InsectEPGSmallTrain | **1.000** | **1.000** | 0.451 | **1.000** | **1.000** | **1.000** | **1.000** | **1.000** | **1.000** | 0.474 |
| ItalyPowerDemand | **0.976** | 0.968 | 0.956 | 0.969 | 0.966 | 0.614 | 0.961 | 0.928 | 0.955 | 0.934 |
| LargeKitchenAppliances | 0.848 | 0.787 | 0.784 | 0.776 | 0.853 | 0.416 | **0.875** | 0.776 | 0.848 | 0.389 |
| Lightning2 | **0.951** | 0.672 | 0.721 | 0.721 | 0.934 | 0.639 | 0.869 | 0.869 | 0.836 | 0.738 |
| Lightning7 | 0.840 | 0.726 | 0.767 | 0.712 | **0.877** | 0.342 | 0.863 | 0.767 | 0.685 | 0.616 |
| Mallat | 0.954 | 0.820 | 0.907 | 0.722 | **0.974** | 0.412 | 0.915 | 0.871 | 0.922 | 0.123 |
| Meat | 0.917 | 0.333 | 0.774 | 0.333 | **0.967** | 0.583 | **0.967** | 0.917 | 0.883 | 0.333 |
| MiddlePhalanxOutlineAgeGroup | **0.669** | 0.610 | 0.617 | 0.597 | 0.662 | 0.577 | 0.636 | 0.643 | 0.630 | 0.578 |
| MiddlePhalanxOutlineCorrect | 0.842 | 0.570 | 0.704 | 0.570 | **0.859** | 0.500 | 0.838 | 0.818 | 0.818 | 0.653 |
| MiddlePhalanxTW | **0.630** | 0.558 | 0.578 | 0.571 | 0.617 | 0.552 | 0.591 | 0.571 | 0.610 | 0.558 |

| Dataset | FreRA (Ours) | best(T) | best(F) | InfoMin | InfoTS | AutoTCL | TS2Vec | TNC | TS-TCC | TF-C |
|---|---|---|---|---|---|---|---|---|---|---|
| MixedShapesRegularTrain | 0.925 | 0.878 | 0.927 | 0.829 | **0.935** | 0.624 | 0.922 | 0.911 | 0.855 | 0.400 |
| MixedShapesSmallTrain | 0.852 | 0.822 | 0.842 | 0.776 | **0.887** | 0.525 | 0.881 | 0.813 | 0.735 | 0.181 |
| MoteStrain | 0.891 | **0.904** | 0.806 | 0.849 | 0.873 | 0.676 | 0.863 | 0.825 | 0.843 | 0.815 |
| OliveOil | 0.800 | 0.400 | 0.752 | 0.400 | **0.933** | 0.600 | 0.900 | 0.833 | 0.800 | 0.400 |
| OSULeaf | **0.909** | 0.678 | 0.705 | 0.554 | 0.760 | 0.384 | 0.876 | 0.723 | 0.723 | 0.467 |
| Phoneme | 0.273 | 0.211 | 0.200 | 0.208 | 0.281 | 0.158 | **0.312** | 0.180 | 0.242 | 0.104 |
| PickupGestureWimoteZ | **0.860** | 0.740 | 0.399 | 0.680 | 0.820 | 0.640 | 0.820 | 0.620 | 0.600 | 0.100 |
| PigCVP | 0.611 | 0.303 | 0.667 | 0.207 | 0.653 | 0.130 | **0.870** | 0.649 | 0.615 | 0.019 |
| PLAID | 0.523 | 0.330 | 0.269 | 0.307 | 0.355 | 0.451 | **0.561** | 0.495 | 0.445 | 0.061 |
| Plane | **1.000** | **1.000** | **1.000** | **1.000** | **1.000** | **1.000** | **1.000** | **1.000** | **1.000** | 0.952 |
| PowerCons | 0.994 | 0.939 | 0.933 | 0.917 | **1.000** | 0.861 | 0.972 | 0.933 | 0.961 | 0.894 |
| ProximalPhalanxOutlineAgeGroup | **0.888** | 0.854 | 0.883 | 0.873 | 0.883 | 0.715 | 0.844 | 0.854 | 0.839 | 0.849 |
| ProximalPhalanxOutlineCorrect | 0.893 | 0.722 | 0.784 | 0.698 | **0.927** | 0.820 | 0.900 | 0.866 | 0.873 | 0.801 |
| ProximalPhalanxTW | **0.849** | 0.678 | 0.800 | 0.780 | 0.844 | 0.771 | 0.824 | 0.810 | 0.800 | 0.795 |
| RefrigerationDevices | 0.597 | 0.549 | 0.533 | 0.501 | **0.624** | 0.360 | 0.589 | 0.565 | 0.563 | 0.299 |
| Rock | 0.700 | 0.480 | 0.480 | 0.500 | **0.760** | 0.400 | 0.700 | 0.580 | 0.600 | 0.280 |
| ScreenType | 0.491 | 0.368 | **0.656** | 0.421 | 0.493 | 0.355 | 0.411 | 0.509 | 0.419 | 0.344 |
| ShakeGestureWiimoteZ | **0.980** | 0.840 | 0.800 | 0.840 | 0.920 | 0.787 | 0.940 | 0.820 | 0.860 | 0.100 |
| ShapeletSim | **1.000** | **1.000** | 0.978 | **1.000** | 0.856 | 0.533 | **1.000** | 0.589 | 0.683 | 0.467 |
| ShapesAll | 0.822 | 0.368 | 0.627 | 0.415 | 0.852 | 0.802 | **0.905** | 0.788 | 0.773 | 0.582 |
| SmoothSubspace | 0.987 | 0.873 | 0.893 | 0.860 | **1.000** | 0.913 | 0.993 | 0.913 | 0.953 | 0.653 |
| SonyAIBORobotSurface2 | **0.957** | 0.815 | 0.867 | 0.807 | 0.953 | 0.769 | 0.890 | 0.834 | 0.907 | 0.846 |
| SonyAIBORobotSurface1 | **0.953** | 0.885 | 0.906 | 0.854 | 0.927 | 0.778 | 0.903 | 0.804 | 0.899 | 0.804 |
| StarLightCurves | **0.973** | 0.874 | 0.964 | 0.891 | **0.973** | 0.849 | 0.971 | 0.968 | 0.967 | 0.855 |
| Strawberry | 0.965 | 0.835 | 0.876 | 0.849 | **0.978** | 0.614 | 0.965 | 0.951 | 0.965 | 0.832 |
| SwedishLeaf | **0.950** | 0.789 | 0.874 | 0.787 | **0.950** | 0.794 | 0.942 | 0.880 | 0.923 | 0.891 |
| Symbols | **0.980** | 0.912 | 0.943 | 0.847 | 0.979 | 0.699 | 0.976 | 0.885 | 0.916 | 0.174 |
| SyntheticControl | **1.000** | 0.990 | 0.980 | 0.997 | **1.000** | 0.880 | 0.997 | **1.000** | 0.990 | 0.760 |
| ToeSegmentation1 | **0.961** | 0.943 | 0.921 | 0.939 | 0.934 | 0.496 | 0.947 | 0.864 | 0.930 | 0.570 |
| ToeSegmentation2 | **0.931** | 0.815 | 0.800 | 0.869 | 0.915 | 0.692 | 0.915 | 0.831 | 0.877 | 0.338 |
| Trace | **1.000** | 0.880 | **1.000** | 0.920 | **1.000** | 0.650 | **1.000** | **1.000** | **1.000** | 0.690 |
| TwoLeadECG | 0.987 | 0.867 | 0.984 | 0.901 | **0.998** | 0.565 | 0.987 | 0.993 | 0.976 | 0.921 |
| TwoPatterns | **1.000** | 0.997 | 0.999 | 0.957 | **1.000** | 0.264 | **1.000** | **1.000** | 0.999 | 0.654 |
| UMD | **1.000** | 0.938 | 0.617 | 0.979 | **1.000** | 0.590 | **1.000** | 0.993 | 0.986 | 0.778 |
| Wafer | 0.996 | 0.960 | 0.956 | 0.959 | **0.998** | 0.921 | **0.998** | 0.994 | 0.994 | 0.994 |
| Wine | 0.833 | 0.500 | 0.500 | 0.500 | **0.963** | 0.500 | 0.889 | 0.759 | 0.778 | 0.500 |
| WordSynonyms | 0.619 | 0.350 | 0.384 | 0.359 | **0.704** | 0.497 | **0.704** | 0.630 | 0.531 | 0.487 |
| Worms | **0.792** | 0.558 | 0.636 | 0.623 | 0.753 | 0.403 | 0.701 | 0.623 | 0.753 | 0.429 |
| WormsTwoClass | 0.831 | 0.753 | 0.714 | 0.714 | **0.857** | 0.558 | 0.805 | 0.727 | 0.753 | 0.584 |
| Yoga | 0.808 | 0.693 | 0.699 | 0.607 | 0.869 | 0.536 | **0.887** | 0.812 | 0.791 | 0.688 |
| Avg. ACC | **0.850** | 0.723 | 0.744 | 0.718 | 0.849 | 0.598 | 0.845 | 0.776 | 0.780 | 0.542 |
| Avg. RANK | 1.940 | 6.320 | 5.750 | 6.470 | **1.930** | 8.420 | 2.670 | 4.810 | 4.670 | 8.330 |

Table 10: The overall classification result of 30 multivariate time series datasets from the UEA archive. The best performance is highlighted in **bold**.

| Dataset | FreRA (ours) | best(T) | best(F) | InfoMin | InfoTS | AutoTCL | TS2Vec | TNC | TS-TCC | TF-C |
|---|---|---|---|---|---|---|---|---|---|---|
| Articulary WordRecognition | **0.990** | 0.887 | 0.947 | 0.913 | 0.987 | 0.983 | 0.987 | 0.973 | 0.953 | 0.467 |
| AtrialFibrillation | **0.467** | 0.400 | 0.333 | 0.267 | 0.200 | **0.467** | 0.200 | 0.133 | 0.267 | 0.040 |
| BasicMotions | **1.000** | **1.000** | **1.000** | **1.000** | 0.975 | **1.000** | 0.975 | 0.975 | **1.000** | 0.475 |
| CharacterTrajectories | 0.991 | 0.953 | 0.976 | 0.990 | 0.974 | 0.976 | **0.995** | 0.967 | 0.985 | 0.090 |
| Cricket | **1.000** | 0.986 | 0.986 | 0.958 | 0.986 | **1.000** | 0.972 | 0.958 | 0.917 | 0.125 |
| DuckDuckGeese | **0.760** | 0.660 | 0.660 | 0.700 | 0.540 | 0.700 | 0.680 | 0.460 | 0.380 | 0.340 |
| Eigen Worms | 0.863 | 0.779 | 0.840 | 0.794 | 0.733 | **0.901** | 0.847 | 0.840 | 0.779 | - |
| Epilepsy | **0.993** | 0.906 | 0.935 | 0.920 | 0.971 | 0.978 | 0.964 | 0.957 | 0.957 | 0.217 |
| ERing | 0.919 | 0.885 | 0.907 | 0.904 | **0.949** | 0.944 | 0.874 | 0.852 | 0.904 | 0.167 |
| EthanolConcentration | 0.323 | 0.297 | 0.262 | 0.243 | 0.281 | **0.354** | 0.308 | 0.297 | 0.285 | 0.247 |
| FaceDetection | **0.581** | 0.564 | 0.521 | 0.560 | 0.534 | **0.581** | 0.501 | 0.536 | 0.544 | 0.502 |
| FingerMovements | 0.610 | 0.530 | 0.500 | 0.500 | 0.630 | **0.640** | 0.480 | 0.470 | 0.460 | 0.510 |
| HandMovementDirection | **0.514** | 0.378 | 0.365 | 0.324 | 0.392 | 0.432 | 0.338 | 0.324 | 0.243 | 0.405 |
| Handwriting | **0.593** | 0.501 | 0.469 | 0.569 | 0.452 | 0.384 | 0.515 | 0.249 | 0.498 | 0.051 |
| Heartbeat | **0.785** | 0.741 | 0.746 | 0.737 | 0.722 | **0.785** | 0.683 | 0.746 | 0.751 | 0.737 |
| Japanese Vowels | 0.965 | 0.938 | 0.938 | 0.938 | **0.984** | **0.984** | **0.984** | 0.978 | 0.930 | 0.135 |
| Libras | **0.911** | 0.761 | 0.822 | 0.800 | 0.883 | 0.833 | 0.867 | 0.817 | 0.822 | 0.067 |
| LSST | 0.494 | 0.393 | 0.391 | 0.473 | 0.591 | 0.554 | 0.537 | **0.595** | 0.474 | 0.314 |
| MotorImagery | 0.550 | 0.530 | 0.540 | 0.530 | **0.630** | 0.570 | 0.510 | 0.500 | 0.610 | 0.500 |
| NATOPS | 0.900 | 0.867 | 0.872 | 0.822 | 0.933 | **0.944** | 0.928 | 0.911 | 0.822 | 0.533 |
| PEMS-SF | 0.746 | 0.653 | 0.671 | 0.699 | 0.751 | **0.838** | 0.682 | 0.699 | 0.734 | 0.312 |
| PenDigits | 0.973 | 0.946 | 0.946 | 0.970 | **0.990** | 0.984 | 0.989 | 0.979 | 0.974 | 0.236 |
| PhonemeSpectra | **0.274** | 0.226 | 0.226 | 0.240 | 0.249 | 0.218 | 0.233 | 0.207 | 0.252 | 0.026 |
| RacketSports | 0.888 | 0.816 | 0.796 | 0.822 | 0.855 | **0.914** | 0.855 | 0.776 | 0.816 | 0.480 |
| SelfRegulationSCP1 | **0.908** | 0.836 | 0.870 | 0.867 | 0.874 | 0.891 | 0.812 | 0.799 | 0.823 | 0.502 |
| SelfRegulationSCP2 | 0.622 | 0.589 | 0.594 | **0.622** | 0.578 | 0.578 | 0.578 | 0.550 | 0.533 | 0.500 |
| SpokenArabicDigits | **0.984** | 0.935 | 0.871 | 0.981 | 0.947 | 0.925 | 0.932 | 0.934 | 0.970 | 0.100 |
| StandWalkJump | **0.667** | 0.400 | 0.333 | 0.333 | 0.467 | 0.533 | 0.467 | 0.400 | 0.333 | 0.333 |
| UWaveGestureLibrary | **0.900** | 0.794 | 0.800 | 0.872 | 0.884 | 0.893 | 0.884 | 0.759 | 0.753 | 0.125 |
| InsectWingbeat | 0.462 | 0.363 | 0.456 | 0.443 | 0.470 | **0.488** | 0.466 | 0.469 | 0.264 | 0.108 |
| Avg. ACC | **0.754** | 0.684 | 0.686 | 0.693 | 0.714 | 0.742 | 0.704 | 0.670 | 0.668 | 0.298 |
| Avg. RANK | **2.133** | 5.967 | 5.800 | 5.500 | 3.967 | 2.600 | 4.967 | 6.433 | 6.033 | 9.276 |

Table 11: The performance of the selected sets of 11 time-domain augmentations on the three HAR datasets. The best performance is highlighted in **bold**, and the second-best performance is underlined. 't_flip', 't_warp', 'perm_jit' and 'jit_scal' are short for `time-flipping`, `time-warping`, `permutation-and-jitter` and `jitter-and-scale`.

| Dataset | FreRA (ours) | jit | scale | negation | perm | shuffling | t_flip | t_warp | resample | rotation | perm_jit | jit_scal |
|---------|--------------|-----|-------|----------|------|-----------|--------|--------|----------|----------|----------|----------|
| UCIHAR | **0.975** | 0.958 | 0.940 | 0.892 | 0.910 | 0.913 | 0.917 | 0.934 | 0.947 | 0.596 | 0.959 | 0.945 |
| MS | **0.982** | 0.930 | 0.914 | 0.813 | 0.927 | 0.910 | 0.915 | 0.925 | 0.956 | 0.887 | 0.948 | 0.915 |
| WISDM | **0.972** | 0.942 | 0.928 | 0.901 | 0.932 | 0.925 | 0.884 | 0.910 | 0.942 | 0.872 | 0.932 | 0.927 |

Table 12: The performance of the selected sets of 5 frequency-domain augmentations on the three HAR datasets. The best performance is highlighted in **bold**, and the second-best performance is underlined.

| Dataset | FreRA (ours) | lpf | hpf | p_shift | ap_p | ap_f |
|---------|--------------|-----|-----|---------|------|------|
| UCIHAR | **0.975** | 0.921 | 0.939 | 0.958 | 0.959 | 0.960 |
| MS | **0.982** | 0.934 | 0.838 | 0.970 | 0.901 | 0.952 |
| WISDM | **0.972** | 0.934 | 0.800 | 0.943 | 0.865 | 0.950 |

