# OpenReview forum: "FreRA: A Frequency-Refined Augmentation for Contrastive Learning on Time Series Classification"
_ICLR.cc/2025/Conference — Submitted to ICLR 2025_

### Official Review · Reviewer_7F2V · 2024-10-28

**Soundness:** 3
**Presentation:** 3
**Contribution:** 3
**Rating:** 5
**Confidence:** 4

**Summary:**

This study introduces Frequency-Refined Augmentation (FreRA), a method designed to overcome limitations in current augmentation strategies for time series classification in contrastive learning. Unlike existing visual-based augmentations, FreRA leverages three key advantages of the frequency domain properties to better preserve the global semantics of time series data. FreRA automatically segregates these components, applying Identity Modification to preserve vital details and Self-adaptive Modification to add variance to less significant parts. Theoretical proofs and empirical evaluations confirm FreRA's superiority, showing it outperforms ten leading baselines across various time series tasks.

**Strengths:**

1. This paper is well-written and well-organized.
2. The methodology is clear and the problem is well-motivated.
3. This is a plug-and-play method that appears to integrate seamlessly with existing contrastive learning frameworks.

**Weaknesses:**

1. Compared to existing works, this work does not exhibit a notable advantage in classification performance.
2. I am somewhat confused about the experimental design for the transfer learning part: 1) Why was SHAR data selected instead of one of the datasets listed in Table 1 (e.g., UCIHAR) to evaluate the transfer capability of the algorithm? 2) Based on the experimental results in Table 2, the performance is lower than that reported in the reference work (Qian et al., 2022). What might be the reasons for this difference?
What did this experiment aim to prove?
3. In the ABLATION STUDIES part, the authors sequentially removed each of the three innovative method components for comparison. From the experimental results, the gains provided by the three modules seem roughly equivalent. If all three modules were removed, what would be the resulting performance? Looking at Table 1 in the paper, the performance of softCLT and FreRA appear quite similar. If FreRA were integrated into softCLT, would there be any gain in performance?

**Questions:**

See weaknesses.

---

> ### Author Response · Authors · 2024-11-22
> **Responses to Reviewer 7F2V [1/2]**
>
> We deeply appreciate the valuable feedback and constructive comments from the reviewer. We would like to address your questions below.
>
> > Q1. Compared to existing works, this work does not exhibit a notable advantage in classification performance.
>
> A1. We would like to highlight the overall performance improvement of our FreRA compared to advanced SOTA methods on the 5 benchmarks in Table 1, as presented in the table below.
>
> |                 | FreRA (Ours) | best(T) | best(F) | InfoMin | InfoTS | AutoTCL | TS2Vec |  TNC  | TS-TCC |  TF-C | SoftCLT |
> |-----------------|:------------:|:-------:|:-------:|:-------:|:------:|:-------:|:------:|:-----:|:------:|:-----:|:-------:|
> | Overall ACC (%) |    **90.66** |   85.28 |   86.20 |   86.16 |  88.56 |   68.22 |  87.84 | 61.66 |  83.52 | 67.30 |   89.52 |
> |   Overall RANK  |     **1.00** |    6.60 |    5.20 |    4.20 |   4.00 |    8.60 |   5.60 |  9.60 |   7.80 |  9.80 |    3.00 |
>
>
> Although the performance gain of FreRA over SoftCLT on some datasets is relatively modest, this is primarily attributed to the use of the soft InfoNCE loss within its TS2Vec framework. For a fair comparison, we present the performance gains of our FreRA and the soft InfoNCE loss on the same TS2Vec framework in the table below (rows 2-3). We observe that FreRA brings larger improvements to the contrastive learning framework.
>
> Moreover, FreRA and the soft InfoNCE loss represent independent improvements to the contrastive learning framework. Due to the **plug-and-play** design of FreRA, these enhancements can be seamlessly integrated. The results in rows 3-4 demonstrate that incorporating FreRA into SoftCLT further introduces additional gains to its performance.
>
> | | CL framework | UCIHAR |MS|WISDM| the factor driving the performance improvements |
> |:-:|:--:|:--:|:--:|:--:|--|
> |1|original TS2Vec|0.959|0.945|0.939||
> |2|FreRA + TS2Vec|0.970 (+0.011)|0.968 (+0.023) |0.957 (+0.018)|FreRA|
> |3|soft InfoNCE + TS2Vec (SoftCLT)|0.961 (+0.002)|0.962 (+0.017) |0.952 (+0.013)|soft InfoNCE|
> |4|FreRA + soft InfoNCE + TS2Vec (FreRA + SoftCLT)|0.969 (+0.008)|0.974 (+0.012)|0.956 (+0.004)|FreRA|
>
> Moreover, in the evaluations of transfer learning and anomaly detection tasks, our FreRA achieves significant improvements. This is evidence that FreRA can capture the inherent semantics of the time series and generalize to unseen data distributions and different downstream tasks.
>
> Lastly, we highlight the significance and contributions of our work as follows:
>
>   - **A novel perspective of designing automatic augmentation from the frequency domain.** We provide a novel perspective to a practical and less-explored problem, automatic augmentation for time series contrastive learning, from the frequency domain. We identify three properties: global, independent, and compact, which advance the view generation and are beneficial for the time series classification task.
>
>    - **A simple yet effective frequency-domain automatic augmentation.** We develop FreRA, a lightweight and unified automatic augmentation method for contrastive representation learning in time series classification tasks. FreRA can be applied in a plug-and-play manner and is jointly optimized with the contrastive learning model.
>
> > Q2. Why was SHAR data selected instead of one of the datasets listed in Table 1 (e.g., UCIHAR) to evaluate the transfer capability of the algorithm?
>
> A2. We selected the SHAR dataset because it presents a **larger domain gap**, as discussed in [1]. This larger domain gap makes the task more difficult and allows us to better assess the generalization and transferability of our method. It also makes our conclusion that FreRA leads to transferable representations more convincing.
>
> > Q3. What might be the reasons for the performance difference with the reference work (Qian et al., 2022)?
>
> A3. The performance difference could be due to the different choices of the encoder and different hyper-parameter search scopes. The experiments in [1] aim to compare performances of various contrastive learning frameworks while our focus is mainly on the performance difference among different augmentations. It is worth noting that, within our work, we apply the same encoder and hyper-parameters search space, which constitute a **fair comparison** with baselines.
>
> [1] Qian, Hangwei, Tian Tian, and Chunyan Miao. "What makes good contrastive learning on small-scale wearable-based tasks?." Proceedings of the 28th ACM SIGKDD conference on knowledge discovery and data mining. 2022.

---

> ### Author Response · Authors · 2024-11-22
> **Responses to Reviewer 7F2V [2/2]**
>
> > Q4. What did the transfer learning experiment aim to prove?
>
> A4. The transfer learning experiment aimed to evaluate whether representation learning with FreRA could effectively transfer useful knowledge to data from unseen domains that have large domain gaps with the training data. The result demonstrates that FreRA achieves **stronger transferability** compared to other baselines. This is attributed to FreRA's ability to infuse variance into the augmented view while leaving critical semantics intact. The encoder thus learns representations that capture the inherent semantics and disregard environment noise. FreRA therefore helps to reduce the domain gap through enhanced augmentation. It further verifies the effectiveness of FreRA.
>
> > Q5. If all three modules were removed, what would be the resulting performance?
>
> A5. If all three modules were removed, the model would downgrade to a vanilla contrastive learning framework where both views are the original input. We present the results averaged over 30 datasets from the UEA archive in the table below. The significant performance drop when all three modules are removed (the fourth column) indicates removing the augmentation leads to collapsed representation learning. To remove all three modules and avoid collapsed training, a simple way is to apply basic augmentations such as pre-defined time-domain and frequency-domain augmentations. The results are presented in the last two columns
>
> |  |FreRA|w/o modification on critical components|w/o modification on noise components| w/o L1 regularization | w/o all three components|best(T)|best (F)|
> |--|:--:|:--:|:--:|:--:|:--:|--|--|
> | Avg. ACC | **0.754** |0.690 (-0.064)|0.695 (-0.059)|0.690 (-0.064| 0.615 (-0.139)|0.684 (-0.070)|0.686 (-0.068)|
>
> > Q6. If FreRA were integrated into softCLT, would there be any gain in performance?
>
> A6. We thank the reviewer for the valuable question. We integrate FreRA into SoftCLT and observe a gain in performances on the three large HAR datasets, as shown in the table below. These results further validate FreRA’s flexibility and effectiveness when applied to other contrastive learning frameworks.
>
> |Dataset|SoftCLT + FreRA|SoftCLT + original augmentation|
> |:--:|:--:|:--:|
> |UCIHAR|**0.969**|0.961|
> |MS|**0.974**|0.962|
> |WISDM|**0.956**|0.952|
>
> We deeply thank the reviewer for raising important questions. We sincerely hope our clarifications above have addressed your concerns and can improve your opinion of our work.

---

> > ### Comment · Reviewer_7F2V · 2024-11-26
> >
> > Thanks to the authors for the efforts to address my comments. My questions have been clarified and I remain my view on this paper.

---

> ### Author Response · Authors · 2024-11-25
> **Kind reminder for discussion**
>
> Dear Reviewer 7F2V,
>
> We would like to express our gratitude for your constructive comments and questions. We look forward to further discussions if you have any remaining questions.
>
> Best,
>
> Paper 6561 Authors

---

### Official Review · Reviewer_GiHe · 2024-11-03

**Soundness:** 3
**Presentation:** 3
**Contribution:** 3
**Rating:** 6
**Confidence:** 4

**Summary:**

The paper introduces a novel augmentation technique designed for time-series contrastive learning by leveraging frequency-domain properties. It utilized the idea of FFT which separates time-series data into critical and non-critical frequency components.

**Strengths:**

The method utilizes the connection between frequency domain knowledge and semantic information to enhance the representation learning. The critical components capture global semantics essential for classification, while non-critical components are used for self-adaptive noise injection, which I found is an interesting link and the authors provide a comprehensive explanation for the motivation.
The authors provide extensive experiments and strong experiment results to demonstrate their method's effectiveness.

**Weaknesses:**

1. At the beginning of the paper, the authors make strong assumptions that existing predefined augmentation methods are primarily adopted from vision and are not specific to time series data. There are already several methods, especially frequency-based augmentation, e.g., TF-C, method design for time series contrastive learning.
2. Since the paper mainly provides the frequency-based augmentation, the motivation study, such as Figure.1 probably should highlight more about whether current frequency-based method can capture the semantics, rather than only focus on the time-domain,

**Questions:**

/

---

> ### Author Response · Authors · 2024-11-22
> **Responses to Reviewer GiHe**
>
> We deeply appreciate the valuable feedback from the reviewer. We would like to address your concerns below.
>
> > Q1. At the beginning of the paper, the authors make strong assumptions that existing predefined augmentation methods are primarily adopted from vision and are not specific to time series data. There are already several methods, especially frequency-based augmentation, e.g., TF-C, method design for time series contrastive learning.
>
> A1. We thank the reviewer for pointing out this important point. We are aware that existing pre-defined augmentations include time-domain and frequency-domain augmentations. Time-domain augmentations are mostly adopted from the vision domain. They often introduce mismatched patterns into the data because they do not account for the intrinsic characteristic of time series. On the other hand, frequency-based augmentations, such as high-pass and low-pass filters, require prior knowledge of the dataset to determine the selection of appropriate augmentation functions. Moreover, other stochastic frequency-domain augmentations, such as the frequency-based augmentation in TF-C, introduce random noise that can interfere with the critical information.
>
> The challenges of existing predefined augmentations can be summarized as follows:
>
>    - They often fail to consider the intrinsic characteristics of time series data, resulting in mismatched patterns due to stochastic perturbations.
>    - Certain augmentations require prior knowledge of the dataset, which is not always accessible in the contrastive learning paradigm.
>    - The wide range of possible augmentation functions requires extensive trials and errors to select the optimal one, making the augmentation process costly and less practical.
>
> The sentence referenced by the reviewer was intended to highlight the problems with existing predefined time-domain augmentations and introduce the novel perspective from the frequency domain. We have revised the sentence to make it more precise and accurate. Moreover, we have included a detailed discussion of the frequency-based augmentations and their problems in Appendix A.2.
>
> > Q2. Since the paper mainly provides the frequency-based augmentation, the motivation study, such as Figure.1 probably should highlight more about whether current frequency-based method can capture the semantics, rather than only focus on the time-domain,
>
> A2. We thank the reviewer for the valuable suggestion. We add the pre-defined frequency-domain augmentation 'amplitude-and-phase-perturbation' denoted as $\mathcal{T}_f(\textsf{x})$ in Figure 1 with explanations updated in our manuscript. It still fails to preserve semantic integrity in the augmented view as we observe from its low mutual information (MI) values. This further explains the limitation of existing augmentations in capturing semantics and highlights the importance of our FreRA.
>
> We deeply thank the reviewer for raising insightful comments. We sincerely hope our clarifications above have addressed your concerns and can improve your opinion of our work.

---

> ### Author Response · Authors · 2024-11-25
> **Kind reminder for discussion**
>
> Dear Reviewer GiHe,
>
> We would like to thank you for your valuable suggestions on enhancing our work. We have provided a response to address your concerns and revised our paper accordingly. We are happy to have further discussions if you have any outstanding concerns or questions.
>
> Best,
>
> Paper 6561 Authors

---

> > ### Comment · Reviewer_GiHe · 2024-11-25
> >
> > Thanks for your clarifications and providing more details, I will keep my score.

---

### Official Review · Reviewer_Qixi · 2024-11-06

**Soundness:** 2
**Presentation:** 2
**Contribution:** 2
**Rating:** 5
**Confidence:** 4

**Summary:**

This paper proposes Frequency-Refined Augmentation (FreRA), an augmentation method for time series contrastive learning on classification tasks. FreRA automatically separates critical and unimportant frequency components, and accordingly proposes Identity Modification and Self-adaptive Modification for different components. It conducts experiments on two benchmark datasets including UCR and UEA archives, as well as 5 large-scale datasets on diverse applications. FreRA outperforms 10 leading baselines on time series classification, anomaly detection, and transfer learning tasks.

**Strengths:**

Data augmentation is an important problem for time series and contrastive learning. This paper investigates this problem and proposes a method from the frequency perspective. The proposed method seems easy to follow and implement. The experiments in this paper are extensive, including many different datasets and tasks. The proposed method outperforms most of the baselines.

**Weaknesses:**

W1: Some important terms in this paper are not clearly defined, such as ‘semantic integrity’ and ‘critical and unimportant frequency components’. Why can we measure semantic integrity using mutual information? Is this consistent with humans’ understanding of semantics? How can we measure the importance of frequency components? What are these critical components critical for?

W2: The overall novelty of the proposed augmentation method is limited. Augmentation from the frequency components is not a new idea for time series. The main difference is a trainable vector s to control the augmentation of different components. It is unclear why s trained from Equation (7) can learn to select critical components automatically.

W3: It is unclear how FreRA achieves both semantic-preserving information and a considerable amount of variance. The authors need to clarify which designs correspond to these two sides respectively.

W4: The self-adaptive modification seems simple and tricky. It only uses the vector s and threshold to select and scale unimportant frequency components. The motivation for scaling these components is unclear.

W5: Compared with some SOTA baselines, such as SoftCLT and InfoTS, the advantage of FreRA is not clear, especially on UEA and UCR datasets. The ablation study is coarse, and some important variants are missing. For example, modifying all (or randomly selected) frequency components and modifying unimportant components randomly.

**Questions:**

Some other questions:

Q1: As this paper focuses on time series classification, why is it also evaluated in anomaly detection?

Q2: Figure 2 is hard to read. The different colors for vector blocks in FreRA are confusing.

---

> ### Author Response · Authors · 2024-11-22
> **Responses to Reviewer Qixi [1/5]**
>
> We thank the reviewer for the time to review our work and offering valuable comments. We address your questions below.
>
> > Q1. clarify important terms
>
> A1. We clarify the important terms as follows.
>    - "Semantic integrity" refers to the **preservation of the meaningful features within the time series that are essential for the downstream classification tasks**, as we explained in lines 65-69 in the revised paper. It is opposed to 'semantic degradation'. In math, it is defined as the state of augmented view $\mathsf{v^\ast}$ when $\text{MI}(\mathsf{v^\ast},\mathsf{y}) = \text{MI}(\mathsf{x},\mathsf{y})$, where $\mathsf{x}$ and $\mathsf{y}$ are the random variables denoting time series sample and the label, and $\text{MI}$ represents mutual information.
>
>    - "Critical frequency components" refers to the frequency components that **contain substantial information related to the downstream classification task**, such as those containing key recurring patterns in the time series. Conversely, "unimportant frequency components" are those that **contribute minimally to classification tasks**, such as noise from the environment.
>
> > Q2. Why can we measure semantic integrity using mutual information? Is this consistent with humans’ understanding of semantics?
>
> A2. From the definition of mutual information, the value of $\text{MI}(\mathsf{v^\ast},\mathsf{y})$ quantifies the amount of information the augmented view $\mathsf{v^\ast}$ can provide about the label. Therefore, it measures the extent to which critical semantics relevant to $\mathsf{y}$ are preserved, which is the definition of semantic integrity as we explained above.
>
> Unlike images and natural language, the semantics of time series are **not intuitively recognizable for human understanding**. In other words, given a time series signal, humans’ understanding of its semantics is ambiguous and vague. Alternatively, by quantitatively measuring the mutual dependency between the time series and its semantics, **the mutual information is superior to humans' understanding in reflecting the completeness of unintuitive semantics**. Moreover, **extensive works [1-3] have applied mutual information to measure the amount of task-relevant semantics information**, which supports its usage in our work.
>
> [1] Oord, Aaron van den, Yazhe Li, and Oriol Vinyals. "Representation learning with contrastive predictive coding." arXiv preprint arXiv:1807.03748 (2018).
>
> [2] Tian, Yonglong, et al. "What makes for good views for contrastive learning?." Advances in neural information processing systems 33 (2020): 6827-6839.
>
> [3] Luo, Dongsheng, et al. "Time series contrastive learning with information-aware augmentations." Proceedings of the AAAI Conference on Artificial Intelligence. Vol. 37. No. 4. 2023.
>
> > A3. How can we measure the importance of frequency components?
>
> A3. Similar to how we quantify the important information in the time domain, we use mutual information to measure the importance of frequency components. Since the frequency components are complex numbers, we calculate MI using only their real parts for compatibility. In Figure 6 of our revised manuscript, we visualize the MI of 3 datasets (Libras, ArticularyWordRecognition, and Epilepsy) by blue-grey bar plots. The important components (those with high MI values with labels) are distributed in low frequencies, middle frequencies, and across multiple frequencies in these datasets, respectively, showing diverse distribution. Despite the diversity, the learned vector $\mathbf{s}$, plotted in orange lines, consistently captures the inherent critical information by learning to assign higher values to the most semantically relevant frequency components. This further verifies the effectiveness and the generalization of FreRA across diverse distributions.
>
> > Q4. What are these critical components critical for?
>
> A4. Intuitively, the critical components are critical for the downstream tasks as they contain task-related information. In practice, preserving the critical components in the augmented view allows the learned representations to have good performance in the downstream classification tasks.

---

> > ### Author Response · Authors · 2024-11-22
> > **Responses to Reviewer Qixi [2/5]**
> >
> > > Q5. The overall novelty of the proposed augmentation method is limited. Augmentation from the frequency components is not a new idea for time series. The main difference is a trainable vector s to control the augmentation of different components.
> >
> > A5. In this work, we address a less-explored but practical and challenging problem, automatic augmentation for time series contrastive learning. We first figure out that **previous augmentation methods fail to preserve semantic integrity** in the augmented view, as illustrated in Figure 1 of our submission, and therefore limits the performance of downstream classification tasks. Based on this observation, we provide a novel perspective from the frequency domain to solve the above problem. The automatic nature of our approach makes it better than previous frequency-based augmentations in three aspects:
> > *  FreRA eliminates the need for extensive parameter tuning and hand-picking, ensuring a more **efficient** augmentation process.
> > * FreRA is deliberately designed to **fully leverage the global, independent, and compact properties of the frequency domain**. It is a unified augmentation that conducts transformation aligned to the inherent semantic distribution of time series instead of stochastic perturbation. The generated views thus **preserve critical semantic information** and are more **effective** for representation learning.
> > * FreRA is designed in a plug-and-use manner, enabling seamless integration with different contrastive learning frameworks. It is consistently beneficial for a variety of contrastive learning frameworks. The empirical comparisons are listed in A4 to reviewer cXW7.
> >
> > > Q6. It is unclear why s trained from Equation (7) can learn to select critical components automatically.
> >
> > A6. We thank the reviewer for raising this important question. The short answer is: $\mathbf{s}$ assigns higher values to critical components. This is the joint result of the maximum agreement objective between the two views and the compactness constraint regularizing the L1-norm of $\mathbf{w}_\text{crit}$.
> >
> > The learning objective for FreRA consists of two components: 1) the contrastive loss and 2) the L1-norm regularization. Optimizing the augmentation with the contrastive loss makes it generate views similar to their corresponding anchors while remaining distinct from other instances, encouraging maximum agreement between the two views. The L1-norm regularization prevents the trivial solution where no changes or minor changes happen to the data, as we explained in lines 371-382 in the revised paper. The overall objective allows **minimal but necessary** critical frequency components to be included in the generated views. As a result, $w_\text{crit}^i$ for critical components are driven toward high values near 1. This differentiates the value of $s_i$, where $w_\text{crit}^i$ is derived from, for critical and unimportant components. The value distinctions in $\mathbf{s}$ allow it to adaptively identify the most semantically relevant components.
> >
> >    Moreover, Figure 6 in our revised manuscript empirically verifies that $s$ learns to assign higher values to the most semantically relevant components and therefore facilitate critical component selection.

---

> ### Author Response · Authors · 2024-11-22
> **Responses to Reviewer Qixi [3/5]**
>
> > Q7. How FreRA achieves both semantic-preserving information and a considerable amount of variance and which designs correspond to these two sides respectively.
>
> A7. Semantic-preserving information and a considerable amount of variance correspond to $\mathbf{w}\_{\text{crit}} \odot x_f$ and $\mathbf{w}\_\text{dist} \odot x_f$ in the Eq. (2) of our manuscript, respectively. They are achieved through the corresponding modification vectors $\mathbf{w}\_{\text{crit}}$ and $\mathbf{w}_\text{dist}$, which are derived from the identity modification module and self-adaptive modification module, respectively. The details are presented below.
>
> 1. Semantic-preserving information is achieved by the **identity modification module on critical frequency components**. Specifically, we learn a lightweight trainable parameter vector $\mathbf{s}$ to capture the inherent semantic distribution in the frequency domain. The semantic-preserving information is preserved by the critical frequency components identified by $\mathbf{s}$.
>
> 2. A considerable amount of variance is achieved by the **self-adaptive modification module on unimportant frequency components**. In the vector $\mathbf{s}$, the value of each element $s_i$ indicates the importance of the $i$-th frequency component $x_f^i$ for the global semantics, as defined in lines 321-322 in the revised paper. Therefore, frequency components with smaller values of $s_i$ are considered unimportant. To infuse variance, we apply perturbation to those unimportant components. Disturbing them does not affect the semantics of the time series because they are independent of the critical frequency components. The selection of unimportant components and the strength of perturbation on each one of them are designed to be adaptive to the input time series dataset. This ensures the amount of variance infused is appropriate.
>
>
> > Q8. The self-adaptive modification seems simple and tricky. It only uses the vector s and threshold to select and scale unimportant frequency components. The motivation for scaling these components is unclear.
>
> A8. The motivation for scaling these components is to infuse variance into the augmented view. Previous studies [3-4] have shown that sufficient variance or task-irrelevant noise can improve the performance of contrastive learning models.
>
>    However, they achieve this by adding random noise in the entire time domain, which introduces variance while inevitably interfering with the critical information. In contrast, FreRA avoids this issue by **selectively** adding variance to the well-separated unimportant frequency components. This ensures the **critical semantics are well preserved**. Moreover, the global property of the frequency components ensures that **all timestamps are altered** with distortion applied only to unimportant components.
>
>    The self-adaptive modification is simple to implement in practice. However, it is **deliberately designed** to **adaptively** add variance to the augmented view. The adaptive nature of the self-adaptive modification is reflected in two aspects:
> 1. Adaptive unimportant component selection: Instead of handpicking a threshold value to separate the unimportant components from the rest, we determine the value with statistical information of the vector $\mathbf{s}$. Specifically, we use the mean value of $\mathbf{s}$ as the threshold and $D = \{i | s_i < \min(0, \frac{1}{F} \sum_{i=1}^{F}{s_i})\}$ to denote the set of unimportant components' indices, as explained in lines 344-350 in the revised paper.
> 2. Adaptive strength of modification of each unimportant component: To adjust the degree of distortion according to the irrelevance of each frequency component, a scaling factor $\delta_s = \frac{1}{\lvert D \rvert} \sum_{i=1}^{F} \mathbb{1}_{\{i \in D\}} \lvert s_i \rvert$ is applied to the unimportant components. This ensures the least important frequency component gets amplified mostly in the distortion step, as explained in lines 351-355 in the revised paper. Its effectiveness has been empirically verified, as detailed in our response to Q10.
>
> Despite the simple but meticulous design, the effectiveness of the self-adaptive modification module has been verified by the result of the ablation study presented in Table 3. It improves the average accuracy of the UEA archive from 0.695 to 0.754, with 0.059 absolute improvement.
>
> [3] Luo, Dongsheng, et al. "Time series contrastive learning with information-aware augmentations." Proceedings of the AAAI Conference on Artificial Intelligence. Vol. 37. No. 4. 2023.
>
> [4] Zheng, Xu, et al. "Parametric Augmentation for Time Series Contrastive Learning." arXiv preprint arXiv:2402.10434 (2024).

---

> ### Author Response · Authors · 2024-11-22
> **Responses to Reviewer Qixi [4/5]**
>
> > Q9. Compared with some SOTA baselines, such as SoftCLT and InfoTS, the advantage of FreRA is not clear, especially on UEA and UCR datasets.
>
> A9. We would like to highlight the overall performance improvement of our FreRA compared to advanced SOTA methods on the 5 benchmarks in Table 1, as presented in the table below.
>
> |                 | FreRA (Ours) | best(T) | best(F) | InfoMin | InfoTS | AutoTCL | TS2Vec |  TNC  | TS-TCC |  TF-C | SoftCLT |
> |-----------------|:------------:|:-------:|:-------:|:-------:|:------:|:-------:|:------:|:-----:|:------:|:-----:|:-------:|
> | Overall ACC (%) |    **90.66** |   85.28 |   86.20 |   86.16 |  88.56 |   68.22 |  87.84 | 61.66 |  83.52 | 67.30 |   89.52 |
> |   Overall RANK  |     **1.00** |    6.60 |    5.20 |    4.20 |   4.00 |    8.60 |   5.60 |  9.60 |   7.80 |  9.80 |    3.00 |
>
>
> Although the performance gain of FreRA over SoftCLT and InfoTS on the UEA and UCR archives is relatively modest, this is primarily attributed to the superior TS2Vec framework they both utilize. To illustrate, we present the baseline performances of TS2Vec and SimCLR in row 1 and row 4 of the table below. We notice that SimCLR is a weaker framework on the UEA and UCR archives. However, integrating FreRA to SimCLR results in significant improvements to the contrastive learning framework, which exceed the performance gains achieved by the augmentation in InfoTS and the soft InfoNCE loss in SoftCLT. This improvement eliminates the performance gap (0.018 for the UEA archive and 0.101 for the UCR archive) caused by the inferior SimCLR framework as compared to TS2Vec.
>
> |   |                   CL framework                   | UEA Archive    |   UCR Archive  | the factor driving the performance improvements |
> |---|:------------------------------------------------:|----------------|:--------------:|-------------------------------------------------|
> | 1 |                  original TS2Vec                 |      0.704     |      0.845     |                                                 |
> | 2 | Information-Aware Augmentation + TS2Vec (InfoTS) | 0.714 (+0.010) | 0.849 (+0.004) | Information-Aware Augmentation                  |
> | 3 |          soft InfoNCE + TS2Vec (SoftCLT)         | 0.751 (+0.047) |  0.850 (+0.005) |                   soft InfoNCE                  |
> | 4 |                  original SimCLR                 |      0.686     |      0.744     |                                                 |
> | 5 |                  FreRA + SimCLR                  | 0.754 (+0.068) | 0.850 (+0.106) |                      FreRA                      |
>
> Moreover, FreRA and the soft InfoNCE loss in SoftCLT represent independent improvements to the contrastive learning framework. Due to the **plug-and-play** design of FreRA, these enhancements can be seamlessly integrated. The results in rows 3-4 demonstrate that incorporating FreRA into SoftCLT further introduces additional gains to its performance.
>
> | |                   CL framework                  |     UCIHAR     |       MS       |      WISDM     | the factor driving the performance improvements |
> |:-:|:-----------------------------------------------:|:--------------:|:--------------:|:--------------:|-------------------------------------------------|
> |1|         SoftCLT         | 0.961  | 0.962  | 0.952  |                                    |
> |2| FreRA + SoftCLT | 0.969 (+0.008) | 0.974 (+0.012) | 0.956 (+0.004) |                      FreRA                      |
>
> Notably, in the evaluations of transfer learning and anomaly detection tasks, our FreRA achieves significant improvements. This is evidence that FreRA can capture the inherent semantics of the time series and generalize to unseen data distributions and different downstream tasks.

---

> ### Author Response · Authors · 2024-11-22
> **Responses to Reviewer Qixi [5/5]**
>
> > Q10. The ablation study is coarse, and some important variants are missing. For example, modifying all (or randomly selected) frequency components and modifying unimportant components randomly.
>
> A10. We thank the valuable advice from the reviewer. We have included a more comprehensive ablation study considering three more variants of infusing variance into the frequency domain:
>
> - Modifying **all** frequency components with stochastic perturbation.
> - Modifying **randomly selected** frequency components with stochastic perturbation.
> - Modifying **unimportant** frequency components with stochastic perturbation.
>
> |          |   FreRA   | perturbation (all) | perturbation (random) | perturbation (unimportant) |
> |----------|:---------:|:------------------:|:---------------------:|:--------------------------:|
> | Avg. ACC | **0.754** |   0.642 (-0.112)   |     0.651 (-0.103)     |       0.703 (-0.051)       |
>
> The results averaged over 30 datasets from the UEA archive are presented in the table above. The performance drop from the third setting highlights the effectiveness of self-adaptive modification applied to unimportant components, as compared to stochastic perturbation. However, randomly disrupting the unimportant components is still better than directly removing them from the generated view ("w/o modification on noise components" in Table 3). The first two settings demonstrate the importance of isolating critical components when introducing noise. Modifying all or random frequency components inevitably interferes with the critical components, which damages the semantic information in the generated views and leads to degraded performance. Randomly perturbing all frequency components results in larger performance drops.
>
> > Q11. As this paper focuses on time series classification, why is it also evaluated in anomaly detection?
>
> A11. In this work, we treat anomaly detection as a 3-class classification problem. Notably, the **class distribution is highly imbalanced** (the class distribution follows 1240:6200:6200) in the anomaly detection dataset. The evaluation results in the anomaly detection tasks further verify the **effectiveness** of our method in a more difficult problem and demonstrate its **generalizability** across different tasks beyond standard time series classification.
>
> > Q12. Figure 2 is hard to read. The different colors for vector blocks in FreRA are confusing.
>
> A12. We thank the reviewer for sharing the feedback on Figure 2. We have made the following amendments in our revised manuscript to make the figure more clear and readable.
> 1. Update the legend to clarify the use of colors for the vector blocks.
> 2. Add explanation to the colored blocks in $\mathbf{w}_\text{dist}$ in the caption.
>
> The use of different colors for the vector $\mathbf{s}$ is intended to indicate that $\mathbf{s}$ learns to assign different levels of importance to the frequency component. The colored blocks in $\mathbf{w}\_\text{dist}$ are to illustrate adaptive distortion strength on unimportant frequency components ($\mathbf{w}\_\text{dist}$ has matching colors with $\mathbf{s}$ on the positions of unimportant components).
>
>
> We deeply thank the reviewer for raising important questions. We sincerely hope our clarifications above have addressed your questions and concerns and can improve your opinion of our work.

---

> ### Author Response · Authors · 2024-11-25
> **Kind reminder for discussion**
>
> Dear Reviewer Qixi,
>
> We would like to express our gratitude for your valuable questions and suggestions. We have provided point-by-point replies to your concerns. We are wondering if our response has properly addressed your concerns. We look forward to discussions with you if you have any outstanding concerns or questions.
>
> Best,
>
> Paper 6561 Authors

---

> ### Comment · Reviewer_Qixi · 2024-11-30
> **Response to rebuttal**
>
> Thank you for the rebuttal. Some of my concerns have been addressed.
>
> I think the technical novelty of the method, which is mainly based on the trainable vector S is still a little limited. Although the vector S can be automatically trained, it still needs to control the trade-off between the contrastive objective and the regularization term to balance huge semantic changes and trivial solutions of no changes. Considering this, the idea of this method seems similar to simply controlling the strengths when using augmentations. Currently, the critical components are a little abstract to me. It may need more analysis to show their properties and why they are critical. Furthermore, using the contrastive objective to learn data augmentation is not reasonable enough for me, as this objective does not clearly encourage meaningful changes in data from the augmentation. It is also unclear why S can be used to adaptively modify unimportant components, considering that it is not directly trained to do this. Besides, this method may need more significant improvements to show its effectiveness. It is not very convincing to claim that FreRA does not outperform SoftCLT clearly because SoftCLT uses TS2Vec.

---

> > ### Author Response · Authors · 2024-12-01
> > **Further Responses to Reviewer Qixi [1/2]**
> >
> > We thank the reviewer for the feedback. We would like to reply to your further concerns below.
> >
> > > Q1. I think the technical novelty of the method, which is mainly based on the trainable vector S is still a little limited
> >
> > A1. We would like to further clarify the technical novelty of our method. While the trainable vector $\mathbf{s}$ is a core component, its value lies not only in its adaptive nature but also in **how it informs and drives the augmentation process**. Specifically, as $\mathbf{s}$ indicates the importance of frequency components, without introducing extra designs, a single $\mathbf{s}$ adeptly guides (1) separation between critical and non-critical components and (2) modifications preserving critical information while introducing variance.
> >
> > We do not hold the belief that a simple yet effective method has limited novelty and contribution. **Despite the lightweight design, FreRA is an effective approach elegantly addressing a nontrivial problem, which can be applied in a plug-and-play manner to a wide range of contrastive learning frameworks.**
> >
> > > Q2. Although the vector S can be automatically trained, it still needs to control the trade-off between the contrastive objective and the regularization term to balance huge semantic changes and trivial solutions of no changes. Considering this, the idea of this method seems similar to simply controlling the strengths when using augmentations.
> >
> > A2. It is worth noting that existing automatic augmentations for time series contrastive learning, i.e., InfoTS and AutoTCL, both rely on trade-off hyper-parameters to balance the loss terms and **this is not unique to FreRA**. Specifically, InfoTS uses **two** trade-off hyperparameters and AutoTCL uses **three**. Compared to them, our FreRA relies only on a **single** trade-off hyper-parameter $\lambda$, which **significantly reduces the cost and complexity of hyper-parameter tuning**.
> >
> > Moreover, as compared to predefined augmentations, FreRA offers a substantial advantage by **alleviating the trials and errors in selecting both the optimal transformation function and the optimal augmentation strength**. This **saves a significant amount of time and resources** in hyper-parameter tuning.
> >
> > Beyond the efficiency in hyper-parameter tuning, FreRA has demonstrated **robust performance towards the selection of hyper-parameter $\lambda$**, as shown in our ablation study on the sensitivity of $\lambda$. Specifically, the results indicate that the performance of FreRA remains stable across different values of $\lambda$ and outperforms the second-best baselines.
> >
> > > Q3. Currently, the critical components are a little abstract to me. It may need more analysis to show their properties and why they are critical.
> >
> > A3. Intuitively, critical components refer to the frequency components that are **most relevant to the labels in the downstream task**. For example, in the Libras dataset, which records hand movements for sign language, the low-frequency components of the recorded signal are critical.  This is because hand movements encoding sign language are smooth and include gradual changes over time, making the low-frequency components most relevant for capturing the global semantics of the gestures. In contrast, high-frequency components mostly represent noise, sensor artifacts, or random fluctuations, which contribute little to the global semantics of hand movements. Similarly, in the Epilepsy dataset recording wrist activities, part of critical information often resides in higher frequency components.  This is because convulsions often happen to people with epilepsy when performing activities, generating high-frequency signals in sensor readings. As a result, the higher frequency components act as a part of critical features.
> >
> > This analysis is supported by our quantitative measurement, i.e., the mutual information between the frequency components and the ground-truth label, shown in the bar plots of Figure 6. For the Libras dataset (on the left), the plot demonstrates that low-frequency components exhibit higher mutual information with the labels, and thus act as critical components for downstream tasks. Conversely, for the Epilepsy dataset (on the right), some higher frequency components demonstrate larger mutual information, indicating their importance for downstream tasks.

---

> > ### Author Response · Authors · 2024-12-01
> > **Further Responses to Reviewer Qixi [2/2]**
> >
> > > Q4. Furthermore, using the contrastive objective to learn data augmentation is not reasonable enough for me, as this objective does not clearly encourage meaningful changes in data from the augmentation.
> >
> > A4. The reviewer may misunderstand how the contrastive objective facilitates our augmentation. Below, we provide clarifications to address the reviewer's misunderstanding.
> >
> > The contrastive objective guides $\mathbf{s}$ in FreRA to preserve semantics in critical components while introducing variance to unimportant components during augmentation, as we defined in Eq. (2), thereby enabling meaningful changes in data.
> >
> > The function of contrastive objective in our augmentation is mainly to facilitate $\mathbf{s}$ to decide the importance of frequency components. Previous work [1] has shown that the contrastive objective is capable of preserving critical information while eliminating random noise. This supports our use of the contrastive objective in learning the importance of frequency components to global semantics with $\mathbf{s}$. Specifically, the contrastive objective encourages higher $s_i$ to critical components so that critical information can be preserved in the augmented view. This has been empirically verified in our visualization of $\mathbf{s}$ from Figure 6. Therefore, the learned $\mathbf{s}$ can conduct meaningful changes from two perspectives:
> >     1. identity modification on critical components with higher $s_i$ to keep semantics information intact
> >     2. self-adaptive modification on unimportant components with lower $s_i$ to introduce variance
> > To summarize, the contrastive objective trains the vector $\mathbf{s}$ to inform and drive the augmentation process.
> >
> > [1] Ji, Wenlong, et al. "The power of contrast for feature learning: A theoretical analysis." Journal of Machine Learning Research 24.330 (2023): 1-78.
> >
> > > Q5. It is also unclear why S can be used to adaptively modify unimportant components, considering that it is not directly trained to do this.
> >
> > A5. The vector $\mathbf{s}$ is trained to decide importance scores for frequency components. (The reviewer can refer to A6 of our earlier response 2/5 and A4 in this response regarding how $\mathbf{s}$ is trained.) This enables $\mathbf{s}$ to effectively distinguish between critical and non-critical components with an adaptive threshold, as explained in lines 342-350 in our revised paper. Additionally, for unimportant components, $\mathbf{s}$ guides the self-adaptive modification module to apply stronger modifications to more irrelevant components, as explained in lines 351-355 in our revised paper.
> >
> > > Q6. Besides, this method may need more significant improvements to show its effectiveness.
> >
> > A6. Our method is a simple yet effective approach and can be applied in a plug-and-play manner. It has demonstrated strong empirical performance on a wide range of datasets in 3 settings, i.e., time series classification, anomaly detection, and transfer learning. Classic contrastive learning frameworks, such as BYOL[2] and SimCLR[3], are also effective approaches with simple designs. Therefore, we believe our current approach is effective enough to address existing challenges and it is less necessary to introduce extra design.
> >
> > [2] Grill, Jean-Bastien, et al. "Bootstrap your own latent-a new approach to self-supervised learning." Advances in neural information processing systems 33 (2020): 21271-21284.
> >
> > [3] Chen, Ting, et al. "A simple framework for contrastive learning of visual representations." International conference on machine learning. PMLR, 2020.
> >
> > > Q7. It is not very convincing to claim that FreRA does not outperform SoftCLT clearly because SoftCLT uses TS2Vec.
> >
> > A7. **The effectiveness of contrastive learning frameworks, such as SimCLR and TS2Vec, DOES make a significant difference across different datasets**, as discussed in prior works [4]. In the second table of our earlier response 4/5, we demonstrate that for the UEA and UCR datasets, SimCLR is less effective than TS2Vec. Consequently, it is challenging for a SimCLR-based approach to be comparable with a TS2Vec-based approach, i.e., SoftCLT. However, **FreRA compensates for the performance gap caused by the contrastive learning framework** and achieves comparable performance with SoftCLT. Moreover, **combining FreRA with SoftCLT further enhances the performance by up to 1.2%**, as shown in the last table of response 4/5.
> >
> > [4] Qian, Hangwei, Tian Tian, and Chunyan Miao. "What makes good contrastive learning on small-scale wearable-based tasks?." Proceedings of the 28th ACM SIGKDD conference on knowledge discovery and data mining. 2022.
> >
> >
> > Lastly, we sincerely hope our above explanations can alleviate the reviewer's concerns about our work. We look forward to any further feedback from the reviewer.

---

### Official Review · Reviewer_cXW7 · 2024-11-08

**Soundness:** 3
**Presentation:** 3
**Contribution:** 2
**Rating:** 5
**Confidence:** 3

**Summary:**

The paper proposes a method, FreRA, to enhance time series classification by contrastive learning and sample augmentations. First, they considered the frequency domain of the time series. FreRA automatically separates the critical and unimportant frequency components. They proposed Identity Modification and Self-adaptive Modification to protect the global semantics in the critical frequency components and inject variance into the unimportant components, respectively. Extensive experimental results on several datasets show that FreRA outperforms existing methods in terms of accuracy.

**Strengths:**

1. Overall, this paper is well-written and easy to follow.
2. The problem studied is significant, and exploring augmentation in time series is novel.
3. Extensive experimental results are promising.

**Weaknesses:**

1. The importance distinction of this method is mostly for the entire time series, and it could be better to compare it with other methods and analyze the theoretical computational complexity.
2. Although frequency methods can improve efficiency, it is unclear whether such methods mainly focus on the low-frequency part and ignore the high-frequency part which is more important for time series prediction.
3. Do the authors consider the dependencies between channels, which is very significant for multivariate time series.
4. The authors claim that FreRA can be benefited by any contrastive learning framework, but only show the results of InfoNCE. What about other CL paradigms, such as SimCLR, etc.?  It could be better to present more sufficient ablation.
5. The experimental results are selected from the highest performances among 11 time-domain augmentations and 5 frequency-domain augmentations. Is this fair enough? There seems to be randomness with such selection strategy.
6. The results of the impacts of hyper-parameters could be moved to the main paper for a better organization.

**Questions:**

See above.

---

> ### Author Response · Authors · 2024-11-22
> **Responses to Reviewer cXW7 [1/3]**
>
> We thank the reviewer for the constructive comments and valuable feedback. We address your concerns below.
>
> > Q1. The importance distinction of this method is mostly for the entire time series, and it could be better to compare it with other methods and analyze the theoretical computational complexity.
>
> A1. We have included comparisons with key baseline methods in the current version, including (1) 11 commonly used handcrafted time-domain augmentations (2) 5 handcrafted frequency-domain augmentations (3) 3 SOTA automatic augmentation for contrastive learning, and (4) 5 SOTA time series contrastive learning frameworks. These baselines also operate on the entire time series and are strong benchmarks for time series classification tasks. **The uniqueness of our FreRA lies in its superior ability to preserve global semantics from the entire time series**, as illustrated in Figure 1 and the results.
>
> The overall computational complexity involves three components as explained below:
>
> 1. The **transformation** involves two parts:
>        - The frequency-domain augmentation involves a Fourier Transform and an inverse Fourier Transform. Implemented by the Fast Fourier Transform (FFT), the computational complexity is $\mathcal{O}(L \log L)$, where $L$ is the sequence length.
>        - Based on the learned $\mathbf{s}$, the identity modification module and the self-adaptive modification module conduct element-wise operations and introduce computation complexity of $\mathcal{O}(F)$.
> 2. **Update of trainable parameters**. The augmentation function is **parameterized** by a lightweight vector $\mathbf{s} \in \mathbb{R}^{F}$, where $F=\lfloor{L/2}\rfloor+1$ is the number of frequency components. The computational complexity of updating $\mathbf{s}$ is $\mathcal{O}(F)$.
> 3. The **auxiliary loss** introduced by the augmentation. $\mathbf{w}_\text{crit}$ is regularized by the L1-norm. The computational complexity introduced by the loss term is $\mathcal{O}(F)$.
>
> The overall computational complexity is dominated by the Fourier Transform and its inverse. Hence, the overall complexity for FreRA can be approximated as $\mathcal{O}(L \log L)$.
>
> In addition, we analyze the computational complexities of the other three SOTA automatic augmentations are present them in the table below. The analysis accounts for per-instance complexity. $B$ and $d$ denote the batch size and feature dimension respectively.
>
> | |FreRA| InfoMin$^+$ |InfoTS|AutoTCL|
> |-|-|-|-|-|
> |transformation|$\mathcal{O}(L\log L)+\mathcal{O}(F)$|$\mathcal{O}(L\log L)+\mathcal{O}(F)$| $\mathcal{O}(7L)$ (7 time-domain augmentations) | $\mathcal{O}(dL) + \mathcal{O}(L)$ (timestamp-level factorization)    |
> | trianable parameter update | $\mathcal{O}(F)$ | $\mathcal{O}(F)$            | $\mathcal{O}(7)$ (weight of 7 candidate augmentations) | $\mathcal{O}(dL)$ (parameters in the factorization and transform functions) |
> | auxiliary loss function    | $\mathcal{O}(F)$ (L1-norm)  | $\mathcal{O}(Bd)$ (InfoNCE) | $\mathcal{O}(Bd)$ (InfoNCE)  | $\mathcal{O}(Bd)$ (MMD)  |
> | Overall                    | $\mathcal{O}(L \log L)$        |      $\mathcal{O}(L \log L + Bd)$           |    $\mathcal{O}(L + Bd)$    |    $\mathcal{O}(L + dL + Bd)$     |
>
> The overall computational complexity of InfoMin$^+$ clearly dominates FreRA. The efficiency of the other three methods depends heavily on the setting of hyper-parameters $B$ and $d$. FreRA consistently achieves competitive performance without imposing significant computational burden, making it a superior option for practical applications.

---

> > ### Author Response · Authors · 2024-11-22
> > **Responses to Reviewer cXW7 [2/3]**
> >
> > > Q2. Although frequency methods can improve efficiency, it is unclear whether such methods mainly focus on the low-frequency part and ignore the high-frequency part which is more important for time series prediction.
> >
> > A2. The distribution of important frequency components **varies across datasets**, instead of having a fixed pattern, such as concentrating on a certain bandwidth. Our method FreRA is designed to **automatically identify the critical components** to preserve global semantic information. We verify this through the following two aspects.
> >
> > * First, we visualize the mutual information (MI) between the frequency components with the label in 3 datasets (Libras, ArticularyWordRecognition, and Epilepsy) by blue-grey bar plots in Figure 6 of our revised manuscript. As shown in the figure, the distribution of important frequency components (those with high MI values with labels) is **dataset-specific**. The important components are distributed in low frequencies, middle frequencies, and across multiple frequencies in these datasets, respectively. The **diversity of the distributions of important components across datasets** makes it unreasonable to directly apply current frequency-domain augmentation such as low- and high-pass filters. Therefore, an **adaptive** augmentation that can learn to identify the critical frequency information becomes practically useful.
> >
> > * Second, to further clarify the effectiveness of FreRA, we visualize the learned vector $\mathbf{s}$ which determines the importance scores of all the frequency components, with the orange line plots in Figure 6. In the plots on all three datasets, despite diverse distributions, $\mathbf{s}$ **consistently captures the inherent critical information** by learning to assign higher values to the most semantically relevant frequency components.
> >
> > > Q3. Do the authors consider the dependencies between channels, which is very significant for multivariate time series.
> >
> > A3. In existing methods, the cross-channel dependencies are usually captured by the encoder. It is in parallel with the augmentation strategy we seek to improve. During the data augmentation process, we conduct uniform transformations within all the channels, which has achieved **empirically competitive results on a wide range of multivariate time series datasets**, including UCIHAR, MS, WISDM, SHAR and the UEA archive. We fully agree that the cross-channel dependencies are important, although current automatic augmentations for multivariate time series have yet to incorporate this piece of information, including our FreRA. We look forward to exploring the incorporation of cross-channel dependency into augmentation strategy design in our future work.

---

> ### Author Response · Authors · 2024-11-22
> **Responses to Reviewer cXW7 [3/3]**
>
> > Q4. The authors claim that FreRA can be benefited by any contrastive learning framework, but only show the results of InfoNCE. What about other CL paradigms, such as SimCLR, etc.? It could be better to present more sufficient ablation.
>
> A4. We thank the reviewer for the insightful comment. In our submission, we apply the architecture of the widely used SimCLR with InfoNCE as the contrastive learning framework. The reason we use InfoNCE instead of NT-Xent, as originally applied in SimCLR, is the better empirical performance, as shown in the table (rows 7-12) below. The same usage has been deployed in [1-2] as well. Moreover, we provide an ablation study evaluating FreRA on alternative contrastive learning frameworks, including TS2Vec, TS-TCC and BYOL, in the Appendix.
>
> Following the suggestions from the reviewer, we have further expanded our evaluation to include additional contrastive learning frameworks: the SimCLR architecture with NT-Xent as contrastive loss functions, as well as an advanced contrastive learning framework SoftCLT. Our current evaluation covers **5 contrastive learning frameworks** and **3 types of contrastive loss functions**. It is worth noting that the contrastive losses used in TS-TCC, TS2Vec, and SoftCLT are different variants of InfoNCE, each with its unique formulation. The results presented below consistently demonstrate that FreRA is a **plug-and-play** method that **consistently and effectively enhances existing contrastive learning frameworks**.
>
> |     | Augmentation + CL framework (contrastive loss) |UCIHAR| MS  |WISDM|
> | -- |:-- |:--:|:--:|:--:|
> | 1  | FreRA + TS2Vec (InfoNCE)   |**0.970**|**0.968**|**0.957**|
> | 2  | original TS2Vec (InfoNCE)  |  0.959  |  0.945  |  0.939  |
> | 3  | FreRA + TS-TCC (InfoNCE)   |**0.944**|**0.959**|**0.962**|
> | 4  | original TS-TCC (InfoNCE)  |  0.924  |  0.915  |  0.889  |
> | 5  | FreRA + SoftCLT (InfoNCE)  |**0.969**|**0.974**|**0.956**|
> | 6  | original SoftCLT (InfoNCE) |  0.961  |  0.962  |  0.952  |
> | 7  | FreRA + SimCLR (InfoNCE)   |**0.975**|**0.982**|**0.972**|
> | 8  | best(T) + SimCLR (InfoNCE) |  0.959  |  0.956  |  0.942  |
> | 9  | best(F) + SimCLR (InfoNCE) |  0.960   |  0.970   |  0.950   |
> | 10 | FreRA + SimCLR (NT-Xent)   |**0.972**|**0.979**|**0.966**|
> | 11 | best(T)+SimCLR (NT-Xent)   |  0.951  |  0.969  |  0.941  |
> | 12 | best(F) + SimCLR (NT-Xent) |  0.955  |  0.965  |  0.952  |
> | 13 | FreRA + BYOL (Cosine Similarity)  |**0.960**|**0.983**|**0.952**|
> | 14 | best(T) + BYOL (Cosine Similarity)|  0.940  |  0.968  |  0.942  |
> | 15 | best(F) + BYOL (Cosine Similarity)|  0.937  |  0.954  |  0.928  |
>
> [1] Yeh, Chun-Hsiao, et al. "Decoupled contrastive learning." European conference on computer vision. Cham: Springer Nature Switzerland, 2022.
>
> [2] Wu, Junkang, et al. "Understanding contrastive learning via distributionally robust optimization." Advances in Neural Information Processing Systems 36 (2024).
>
> > Q5. The experimental results are selected from the highest performances among 11 time-domain augmentations and 5 frequency-domain augmentations. Is this fair enough? There seems to be randomness with such selection strategy.
>
> A5. The highest performances we present are the **upper limits of pre-defined augmentations**. Such a selection strategy, listing only the highest results, also intends to ensure a **simple and concise** tabular presentation **without randomness**.
>
> The selection of the optimal pre-defined augmentation is data-specific and there is no unified pre-defined augmentation that works well on all the datasets. Even if we report the best empirical results of the optimal augmentations selected from exhaustive trials and errors on each dataset, they are still worse than the results of our FreRA. In this context, the comparison is fair enough, as the selection strategy aims to find out the augmentations that empirically best suit the given dataset. This comparison again highlights the effectiveness of FreRA.
>
> > Q6. The results of the impacts of hyper-parameters could be moved to the main paper for a better organization.
>
> A6. We thank the reviewer for the suggestion. We have included the ablation study on the impacts of hyper-parameters in the main paper accordingly. Due to the space limit, the plot and the detailed analysis remain in the Appendix, but the main conclusions of this ablation study are accessible from the main paper.
>
> We deeply thank the reviewer for raising insightful comments. We sincerely hope our clarifications above have addressed your concerns and can improve your opinion of our work.

---

> ### Author Response · Authors · 2024-11-25
> **Kind reminder for discussion**
>
> Dear Reviewer cXW7,
>
> We would like to thank you again for providing valuable feedback and constructive suggestions. Please kindly let us know if our response has addressed your questions. We look forward to more discussions if you have further comments.
>
> Best,
>
> Paper 6561 Authors

---

### Author Response · Authors · 2024-11-26
**General Response**

Dear ACs and Reviewers,

We sincerely appreciate your time and effort in reviewing our work and providing constructive feedback. We would like to 1) express our gratitude for reviewers’ recognition of our work, and 2) highlight the major modifications made in our revised paper.

**We thank the reviewers for recognizing and appreciating the advantages of our work.**

* Investigating automatic augmentation in time series is **significant, novel and well-motivated**. [cXW7,Qixi,7F2V,GiHe]
* The proposed methodology is **clear, interesting, novel, and easy to follow and implement**.  [7F2V,GiHe,Qixi]
* The proposed FreRA is a **plug-and-play** method that can be integrated seamlessly with existing contrastive learning frameworks.[7F2V]
* **Extensive experimental results** are **promsing or strong**. [cXW7, Qixi, GiHe]
* The paper is **well-written, well-organized and easy to follow**. [cXW7,7F2V]

Besides the response to each reviewer, we would like to summarize the **major modifications made in our revised paper (highlighted in blue)**:

1. **More evaluations on different contrastive learning frameworks.** We expand the evaluation of FreRA on 3 additional contrastive learning frameworks. The results reported in Table 5 and Table 6 of Appendix A.9 demonstrate that FreRA is a **plug-and-play** method and it **consistently and effectively enhances existing contrastive learning frameworks**. [cXW7,7F2V]

2. **Visualization and analysis on the learned vector $\mathbf{s}$.** We visualize the learned vector $\mathbf{s}$ and demonstrate its ability to **capture the diverse distributions of critical semantics** across three different datasets. The visualization is provided in Figure 6, with explanations in Appendix A.9. [cXW7,Qixi]

3. **More discussion and visualization on predefined frequency-domain augmentations.**

    * We revise the statements regarding our motivation for investigating the frequency domain in the abstract to make it precise and accurate. Additionally, we include a detailed discussion on frequency-based augmentations and their limitations in Appendix A.2. [GiHe]
    * We include the predefined frequency-domain augmentation in Figure 1 to illustrate it also causes undermined semantics in the generated views. Corresponding explanations are updated accordingly. [GiHe]

4. **Presentation enhancements.**

    * We update the legend and caption in Figure 2 to clarify the use of colors for the vector blocks. [Qixi]
    * We move the ablation study on the impact of hyper-parameter from the appendix to the main paper for better organization. [cXW7]


Thank you once again for taking the precious time to review our work. We would be delighted to engage in further discussions if you have any remaining questions or concerns.

Best regards,

Paper 6561 Authors

---

### Meta-Review · Area_Chair_ggEJ · 2024-12-22

**Metareview:**

This paper presents a new augmentation method named Frequency-Refined Augmentation (FreRA) for time series contrastive learning and classification. Reviewers agreed that the paper is well written and easy to follow, the method is well motivated, and the experiments are extensive. Meanwhile, reviewers pointed out that there are still some limitations regarding technical contribution, details on methodology, experiments, novelty, etc. Although some of these concerns have been addressed during the rebuttal and discussion stage, some issues still remain. For instance, the novelty of the proposed method is not significant, and the advantages of FreRA over existing work are not sufficiently justified. Overall, this is a borderline paper.

**Additional Comments On Reviewer Discussion:**

Reviewers raised many concerns regarding technical contribution, details on methodology, experiments, novelty, etc. The authors provided detailed responses and additional results, which have addressed some of these concerns. However, during the post-rebuttal discussions, the concerns on novelty and technical contributions still remain.

---

### Decision · Program_Chairs · 2025-01-22

Reject